# LLM ROUTING WITH DUELING FEEDBACK

## ABSTRACT

We study LLM routing, the problem of selecting the best model for each query while balancing user satisfaction, model expertise, and inference cost. We formulate routing as contextual dueling bandits, learning from pairwise preference feedback rather than absolute scores, thereby yielding label-efficient and dynamic adaptation. Building on this formulation, we introduce Category-Calibrated Fine-Tuning (CCFT), a representation-learning method that derives model embeddings from offline data using contrastive fine-tuning with categorical weighting. These embeddings enable the practical instantiation of Feel-Good Thompson Sampling for Contextual Dueling Bandits (FGTS.CDB), a theoretically grounded posterior-sampling algorithm. We propose four variants of the categorical weighting that explicitly integrate model quality and cost, and we empirically evaluate the proposed methods on the RouterBench and MixInstruct datasets. Across both benchmarks, our methods achieve lower cumulative regret and faster convergence, with better robustness and performance-cost balance than strong baselines built with a general-purpose OpenAI embedding model.

## 1    INTRODUCTION

The potential of large language models (LLMs) is so great that they have become a necessary part of daily life, with applications ranging from office assistance and fashion/dining suggestions to entertainment. LLM routing refers to a problem of dynamically selecting the most suitable LLM from a set of candidates for each query in a sequence of questions. Before the emergence of a universally dominant and affordable foundation model, routing is important because the choice of LLM must align with user traits, model expertise, and cost. To balance these three key factors, cascading algorithms such as FrugalGPT (Chen et al., 2024) and AutoMix (Aggarwal et al., 2024) were first proposed. The idea is to query a cheaper model first and advance the query to a more expensive one if the current response is unlikely to meet the user's expectation.

A drawback of cascading is the accumulated cost and latency caused by calling multiple LLM candidates to generate the final response. To avoid this, supervised routing methods (Shnitzer et al., 2023; Lu et al., 2024; Ding et al., 2024; Hu et al., 2024; Srivatsa et al., 2024) were proposed. In general, a supervised router reduces the latency by applying a classification or a regression prediction before the LLM query. The supervised approach has evolved into several branches. A branch studied ensembles (Jiang et al., 2023; Maurya et al., 2025; Zhang et al., 2025b;a), which allows the agent to select a subset or fuse answers. Another branch focused on cost-aware model assignment (Šakota et al., 2024; Hu et al., 2024; Liu et al., 2024) to balance cost and performance. There are also works that combine representation learning (Feng et al., 2025b; Zhuang et al., 2025) to strengthen the user and model's semantic information before training.

For supervised routing methods, having abundant real-valued annotations with high-quality label information is critical for successful classification or regression training. Unfortunately, such a requirement is often unrealistic in the context of LLM routing. In some cases, users are either reluctant or unmotivated to provide feedback. They may also be unable to quantify their satisfaction, especially when dealing with open-ended questions or when they lack the ability to verify the correctness of the LLM's response. To ease the annotation burden, some efforts focused on weak supervision (Sugiyama et al., 2022; Chiang & Sugiyama, 2025), which only collects binary feedback such as like/dislike or pairwise comparison (Ong et al., 2025; Zhao et al., 2024; Wang et al., 2025). The advantage is that one-click feedback is user-friendly, and it is more confident to say response A is better than B than to assign response A a score out of ten. The reason that makes a weakly supervised

approach appealing is that, as shown by the referred papers, the binary feedback can be translated into a model ranking or an estimate of the labeling function.

In addition to the challenge of annotation, adaptivity remains a key challenge to be addressed in developing a usable routing system. Shifts in query distributions, such as changes in trending topics like fashion or temporal variations between work hours and leisure time, introduce non-stationary conditions. Moreover, new LLMs and benchmarks are continuously introduced, resulting in a constantly evolving environment for routing. Because of their static nature, supervised learning-based routing policies struggle to address multiple adaptation challenges simultaneously. Addressing such challenges in a dynamic environment is a key motivation for adopting online learning approaches for LLM routing, as these offer the ability to continuously learn and optimize the routing policy in real time. The online algorithms adopted in prior work can be categorized into three classes: multi-armed bandits (Nguyen et al., 2025; Dai et al., 2024; Li, 2025), contextual bandits (Wang et al., 2025), and reinforcement learning (Sikeridis et al., 2025).

To build a practical routing system that fits the various requests, multiple challenges should be addressed simultaneously. However, we notice that little effort has been made to jointly solve the challenges of adaptivity and weak supervision, even if the community has already made significant progress in respective directions. To the best of our knowledge, Wang et al. (2025) is the only attempt to address adaptivity and pointwise feedback (e.g., like/dislike) at the same time. Therefore, this paper focuses on investigating LLM routing under a stochastic bandit setting, which captures the dynamics of a changing environment, and operates under weak supervision in the form of pairwise preference feedback (e.g., response A is preferred over response B). The advantages of the project are threefold: First, we introduce the Feel-Good Thompson Sampling for Contextual Dueling Bandits (FGTS.CDB) algorithm (Li et al., 2024) as the core module, which naturally integrates weak supervision (dueling feedback) and adaptive learning (bandit algorithm) in both input and learning design, expanding methodological options for future research. Second, FGTS.CDB is theoretically grounded, providing a clear explanation of how binary feedback relates to a utility function shaped by user satisfaction, model expertise, and cost. Third, it offers a platform to analyze its strengths and limitations, enabling development of a practical LLM router that does not rely on high-quality annotations and remains robust in dynamic environments.

The paper's contributions are summarized as follows.

- We propose Category-Calibrated Fine-Tuning (CCFT), a generic embedding strategy to encode LLM expertise. The feature function from CCFT enables the first trainable contextual dueling learner for LLM routing that operates purely on binary preference feedback, without using per-model scalar performance labels.

- Strong evidence for the efficacy of the proposed strategy is provided by experiments on two real-world datasets, RouterBench (Hu et al., 2024) and MixInstruct (Jiang et al., 2023). Four CCFT variants are implemented and evaluated, and the cumulative regret curves show convergence to the optimal strategy, selecting the best-matching model for each query.

- The proposed methods demonstrate robust generalization on the unseen benchmark and achieve a balance between performance and cost. They incorporate common practices, including prompting, embedding model fine-tuning, and the use of both open-source and black-box text embedding models. Therefore, the experiments contribute to the accumulation of substantial knowledge and expertise in addressing LLM routing challenges.

## 2 RELATED WORK

**LLM selection strategies** LLM selection can be organized along two axes: how candidates are queried and what learning signal is used. On the querying side, cascading systems (Chen et al., 2024; Aggarwal et al., 2024; Narasimhan et al., 2025; Chuang et al., 2025) issue a sequence of calls, starting from a cheap model and escalating to stronger ones until a confidence or quality threshold is met. In contrast, one-shot routers predict a single or two target model(s) before inference. One-shot routing is preferable when latency must remain small or when a diverse pool of models with complementary domain strengths is available (Jiang et al., 2023) and we wish to select one (or two, for preference feedback). Within one-shot routing, offline methods train a classifier or regressor on a fixed labeled set to map queries to models (e.g., Shnitzer et al., 2023; Lu et al., 2024; Ding et al., 2024; Hu et al., 2024; Srivatsa et al., 2024; Jitkrittum et al., 2025). Online methods instead adapt

the routing policy on the fly using bandits or reinforcement learning to cope with distribution shift and evolving model pools (e.g., Nguyen et al., 2025; Dai et al., 2024; Li, 2025; Wang et al., 2025; Sikeridis et al., 2025). Note that in our setting, the algorithm is formulated to output two LLMs but the same LLM can be chosen twice. In such cases, we naturally only require a single call similar to a successful cascading system.

On the signal side, many routers rely on pointwise supervision, i.e., correct/incorrect or scalar ratings, while others leverage preference (pairwise) feedback that compares two candidates, which can be easier to elicit (Ong et al., 2025; Zhao et al., 2024; Wang et al., 2025). Our work lies in the online, one-shot setting with preference signals: we model routing as contextual dueling bandits and instantiate a Thompson-sampling-style learner that updates from pairwise comparisons while balancing performance and cost. Most of previous routers (Jitkrittum et al., 2025; Pulishetty et al., 2025; Somerstep et al., 2025; Feng et al., 2025a; Zhuang et al., 2025) rely on pointwise supervision such as correctness labels or scalar scores for each candidate model. By contrast, our algorithm is trained only from pairwise preferences between two sampled models per query and never observes absolute performance labels.[1]

**Contextual Dueling Bandits and Feel-Good Thompson Sampling**  The contextual bandit problem extends the classical multi-armed bandit setting by leveraging side information (Langford & Zhang, 2007). It has found widespread applications in areas such as online advertising, recommender systems, and mobile health (Li et al., 2010; Agarwal et al., 2016; Tewari & Murphy, 2017). A widely used and empirically effective class of algorithms for contextual bandits is Thompson Sampling (TS) (Thompson, 1933), known for its strong empirical performance Chapelle & Li (2011); Osband & Van Roy (2017). Research on contextual dueling bandits has taken several algorithmic and theoretical directions. Kumagai (2017) analyzed dueling bandits with a continuous action space and, under strong convexity and smoothness, established dimension-free regret guarantees. Building on preference models, Bengs et al. (2022) introduced the CoLSTIM algorithm for stochastic contextual dueling bandits under linear stochastic transitivity, providing learning guarantees tailored to this structure. Recently, Di et al. (2024) proposed VACDB, an action-elimination-based method that achieves tighter, variance-dependent regret bounds for contextual settings.

Feel-Good Thompson Sampling (FGTS) was proposed to reconcile TS's strong empirical performance with frequentist-style guarantees (Zhang, 2022). Fan & Gu (2023) offered a unified analysis framework showing how FGTS yields robust guarantees across several linear contextual bandit variants. The FGTS idea has also been extended to reinforcement learning, e.g., Model-based Optimistic Posterior Sampling (MOPS) for Markov decision processes (Agarwal & Zhang, 2022). To the best of our knowledge, our work is the first to apply FGTS to LLM routing, connecting preference-based bandit principles with practical model-selection pipelines.

## 3 BACKGROUND

The contextual dueling bandit problem can be seen as a repeated game between a bandit algorithm and an environment for $T$ rounds. In each round $t = 1, 2, \ldots T$, the algorithm observes a contextual vector $x_t$ from the environment. Then, the algorithm selects two actions $a_t^1, a_t^2 \in \mathcal{A} = \{a_k\}_{k=1}^K$ in response to the environment. After presenting the responses, the algorithm observes a preference feedback $y_t$. The performance of the algorithm is measured by its cumulative regret

$$\text{Regret}(T) := \sum_{t=1}^{T} \left[ r^*(x_t, a_t^*) - \frac{r^*(x_t, a_t^1) + r^*(x_t, a_t^2)}{2} \right], \tag{1}$$

where $r^*(x, a)$ is the utility function and $a_t^* = \arg\max_{a \in \mathcal{A}} r^*(x_t, a)$ is the best action for input $x_t$.

The setting fits seamlessly to the LLM routing problem studied in this paper if we view $x_t$ as the query embedding, $a_t^1$ and $a_t^2$ as two LLMs, $y_t$ as the preference feedback, and the $r^*(x, a)$ as the function balancing the user satisfaction and model score[2] that we want to optimize. Minimizing $\text{Regret}(T)$ precisely captures the goal of routing: identifying the optimal LLM $a_t^*$ at each round, rather than committing to a single fixed $a^*$ across all rounds. The connection between the weak

---

[1]This difference in supervision regime is the main reason why existing supervised routers cannot be used as drop-in baselines for our setting, in § 5.

[2]The model score is computed based on LLM performance, cost, latency, and other relevant factors.

supervision of preference feedback $y$ and the ideal supervision provided by $r^*(x, a)$ is captured by the Bradley–Terry–Luce (BTL) model (Hunter, 2004; Luce, 2005): Given query $x$ and two LLMs $a^1$ and $a^2$, the probability of observing $a^1$ is preferred over $a^2$ (i.e., $y = 1$) is

$$\mathbb{P}(y = 1 \mid x, a^1, a^2) = \exp\left(-\sigma(r^*(x, a^1) - r^*(x, a^2))\right),$$

where $\sigma(z) = \log(1 + \exp(-z))^3$.

Assuming the linear reward model $r^*(x, a) = \langle \theta^*, \phi(x, a) \rangle$, Li et al. (2024) proposed Alg. 1 and proved that it achieves $\mathbb{E}[\text{Regret}(T)] = \widetilde{\mathcal{O}}(d\sqrt{T})$, where $d$ is the dimension of $\theta$. FGTS.CDB learns the LLM selection strategies, $\theta_t^1$ and $\theta_t^2$, in an online fashion. Its success hinges on the posterior $p^j(\theta \mid S_{t-1})$ defined by the likelihood function

$$L^j(\theta, x, a^1, a^2, y) = \eta\,\sigma\left(y\langle\theta, \phi(x, a^1) - \phi(x, a^2)\rangle\right) - \mu \max_{\tilde{a} \in \mathcal{A}}\langle\theta, \phi(x, \tilde{a}) - \phi(x, a^{3-j})\rangle. \quad (2)$$

The intuition behind $L^j(\cdot)$ is as follows: if $\theta$ aligns well with the preference feedback $y$, that is, when $y\langle\theta, \phi(x, a^1) - \phi(x, a^2)\rangle$ is positive and large, then $\theta$ is more likely to be chosen. The second term in the likelihood function serves as a "feelgood" component, encouraging the selection of a $\theta$ that outperforms past selections made by the other selection strategy[4]. As time $t$ progresses, the observation history $S_t$ accumulates, allowing FGTS to dynamically adjust its samples $\theta_{t+1}^1$ and $\theta_{t+1}^2$ accordingly. In an evolving environment, any changes are reflected in $S_t$, and adaptivity is inherently ensured by the bandit algorithm.

---

**Algorithm 1** FGTS.CDB

1: Given action space $\mathcal{A}$ and hyperparameters $\eta, \mu$. Initialize $S_0 = \varnothing$.
2: **for** $t = 1, \ldots, T$ **do**
3:     Receive query $x_t$.
4:     **for** $j = 1, 2$ **do**
5:         Sample model parameter $\theta_t^j$ from posterior

$$p^j(\theta \mid S_{t-1}) \propto \exp\left(-\sum_{i=1}^{t-1} L^j(\theta, x_i, a_i^1, a_i^2, y_i)\right) p_0(\theta)$$

6:         Select LLM $a_t^j = \arg\max_{a \in \mathcal{A}} \langle\theta_t^j, \phi(x_t, a)\rangle$ to generate the response.
7:     Receive preference feedback $y_t$.
8:     Update history $S_t \leftarrow S_{t-1} \cup \{(x_t, a_t^1, a_t^2, y_t)\}$.

---

Being theoretically oriented, Li et al. (2024) assumes that the feature function $\phi(x, a)$ is given and perfect. However, this assumption generally does not hold in real-world applications, as shown in the following section.

## 4 A GENERIC REPRESENTATION LEARNING STRATEGY

In § 4.1, we demonstrate how an imperfect $\phi(x, a)$ can hinder learning and highlight a key obstacle in applying FGTS.CDB to the routing problem. Subsequently, in § 4.2, we introduce our methods for constructing feature functions that enable FGTS.CDB to function effectively in practical LLM routing scenarios.

### 4.1 THE FAILURE OF NAIVE IMPLEMENTATIONS

This subsection highlights the practical challenges involved in designing an effective feature function. We construct synthetic simulations based on the MMLU dataset (Hendrycks et al., 2021). To this end, we use OpenAI's `text-embedding-3-large` model to implement two straightforward embedding methods: `OpenAItext_prompt`, which uses prompting, and `OpenAItext_mean`, which relies on averaging embeddings. Their experimental details are defer to App. A.1.

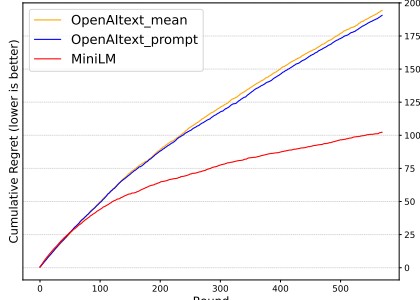

Figure 1: Failed versus successful examples.

---

[3]A more general setting accepting more than two candidates is called the Plackett-Luce (PL) model (Azari Soufiani et al., 2014; Khetan & Oh, 2016; Ren et al., 2018).

[4]Specifically, note that, $a^{3-j} = a^2$ for $j = 1$ and $a^{3-j} = a^1$ for $j = 2$.

From Fig. 1, we see that the slopes of `OpenAItext_mean` and `OpenAItext_prompt` almost do not change with rounds. This means the regret keeps accumulating, and hence the learning does not progress a lot. In contrast, the red curve resembles the learning behavior we want. The slope of the red curve reduces as the rounds increase. This results in small cumulative regret, reflecting that the learning agent is making fewer and fewer mispredictions and converging the behavior toward the best routing policy. In the next subsection, we propose a generic strategy to construct model embeddings that aims to achieve behavior similar to the red curve, and we test its implementations on real-world datasets in experiments.

## 4.2 THE PROPOSED METHOD

Note that by design, $\phi(x_t, a_k)$ combines the information from the user side, $x_t$, and the information from the LLM side, $a_k$. Since the text models we selected are well-recognized sentence encoders, the failure cases discussed in § 4.1 are most likely due to careless design of the model embeddings. Therefore, it is imperative to ensure that $a_k$ accurately captures the connection between the prospective queries and the LLM's expertise. In this section, we introduce *Category-Calibrated Fine-Tuning (CCFT)*, a generic strategy for constructing $a_k$ through contrastive fine-tuning combined with categorical weighting. Our working hypothesis is that an LLM's expertise is characterized by the categories (or benchmarks) on which it performs well, and that the semantics of each category are determined by the queries belonging to it. CCFT operationalizes this hypothesis in three steps: (i) we learn question embeddings that form tight clusters within each category via contrastive fine-tuning, (ii) we average the resulting question embeddings within each category to obtain category embeddings, and (iii) we combine these category embeddings into a single LLM embedding using category-level performance metadata.

Suppose the queries can be divided into $M$ categories. The 2D t-SNE plot of the failure examples in Fig. 5 shows that the text model tends to cluster queries from the same category. It provides contrastive fine-tuning a good starting point. Thus, we fine-tune the text model using a small offline query set that is disjoint from the online testing set. The fine-tuning is applied only to the question encoder, using a cosine-similarity contrastive loss over positive and negative query pairs constructed from category or benchmark labels, so that queries from the same category form denser clusters and separate more clearly from other categories (see App. A.1 for the exact objective and training details). No additional fine-tuning is performed when constructing category or model embeddings. Then, for each category $m$, we compute the category embedding $\xi_m$ by averaging the embeddings of offline queries belonging to that category, as generated by the fine-tuned text model. Note that the category embedding $\xi_m$ is not the model embedding $a_k$, for which an additional step is required, as described next.

To capture the unique characteristics of each LLM, we assume that every LLM is associated with a distinct Kiviat diagram, representing its areas of expertise. Under this assumption, a natural approach is to compute a weighted combination of the category embeddings, where the weights are derived from the LLM's scores on its Kiviat diagram. We refer to this mechanism as "categorical weighting". In particular, we propose the following four weighting methods. Denote $M$ category embeddings $(\xi_1, \xi_2, \ldots, \xi_M) := \xi$. For each LLM, let $(s_{k,1}, s_{k,2}, \ldots, s_{k,M})^\top := s_k$ be the score vector over the categories. The first weighting method, coined "`perf`", defines the model embedding as

$$a_k = \xi \operatorname{softmax}(s_k), \tag{3}$$

in which $s_{k,m}$ is the model performance on category $m$. If the score $s_{k,m}$ is a function of model performance and model cost, we coin the resulting $a_k$ "`perf_cost`". Note that `perf` and `perf_cost` take all score values into account. In reality, a common scenario involves a few strong models that specialize in different categories, alongside several weaker models that perform comparably within specific domains. In such a case, we could weight an LLM only on categories it is good at, and leave the other categories handled by other LLMs. Formalizing this idea, we propose "`excel_perf_cost`" as

$$a_k = \xi \operatorname{softmax}\left(\operatorname{top}^{(\tau)}(s_k)\right), \tag{4}$$

and "`excel_mask`" as

$$a_k = \xi \frac{\operatorname{mask}^{(\tau)}(s_k)}{\tau}. \tag{5}$$

Let $\tau \in \{1, 2, \ldots, K\}$, and let $s_{(\tau),m}$ denote the $\tau$-th largest value in $\{s_{1,m}, s_{2,m}, \ldots, s_{K,m}\}$[5]. Function $\text{top}^{(\tau)}(s_k)$ produces a real-valued vector with $m$-th entry $\text{top}^{(\tau)}(s_k)_m = s_{k,m}\mathbf{1}[s_{k,m} \geq s_{(\tau),m}]$. Similarly, function $\text{mask}^{(\tau)}(s_k)$ produces a binary vector with $m$-th entry $\text{mask}^{(\tau)}(s_k)_m = \mathbf{1}[s_{k,m} \geq s_{(\tau),m}]$. Now, we have proposed four variants for computing an LLM embedding. Under the linear reward model $r^*(x, a) = \langle \theta^*, \phi(x, a) \rangle$ introduced in § 3, this construction cleanly separates LLM expertise from user preference: $\phi(x, a)$ encodes how well model $a$ is suited to a query $x$ based on category-level structure and offline metadata, while $\theta^*$ captures user-specific trade-offs between performance and cost. In our implementation $\phi$ is fixed after the offline CCFT step, and the online bandit learner updates its posterior over $\theta$ using only binary preference feedback $y$ through the likelihood in (2). This separation is what allows us to rely on very small offline question sets while still adapting online to user preferences.

**Categorical Weighting without Scores**   Note that the weighting mechanisms (3), (4), and (5) require score information, which might not always be the case. Next, we show that we can still perform a way of categorical weighting under a mild condition.

The term *label* here refers to the index of the LLM that is most preferred for a query (for example, the model that wins the majority of pairwise comparisons for that query); it is distinct from the *category* or benchmark that the query comes from[6]. We do not require explicit category labels in the dataset, and categories can be interpreted as latent subpopulations of queries.

Suppose that for each query we know which LLM is currently the best match, and we record this as its label $k \in \{1, 2, \ldots, K\}$. Thus, we use $f_{km}$ to denote the proportion of queries in (latent) category $m$ whose label is $k$.

We assume the offline data is generated as follows: From each category $m$, a set of query embeddings $\mathcal{Q}_m$ of size $n$ is sampled. The offline data is $\{\mathcal{G}_k\}_{k=1}^K$, formed by regrouping $\{\mathcal{Q}_m\}_{m=1}^M$ according to the labels. Given the offline data, we propose to compute the model embedding, for each $k \in \{1, 2, \ldots, K\}$, as

$$a_k = \sum_{q \in \mathcal{G}_k} q/|\mathcal{G}_k|. \tag{6}$$

The following proposition justifies the proposed mechanism; the proof is deferred to App. A.2, where we also provide a simple two-category example to illustrate its intuition.

**Proposition 1.** *Let $f_{km}$, $\{\mathcal{Q}_m\}_{m=1}^M$, and $\{\mathcal{G}_k\}_{k=1}^K$ be defined as above. Let $\mathbb{E}[Q_m]$ denote the expected embedding of queries in category $m$. Assume the embedding distribution within category $m$ is independent of label $k$[7]. Then, for each $k \in \{1, 2, \ldots, K\}$, the average embedding (6) is an unbiased estimate for $\sum_{m=1}^M \frac{f_{km}}{\sum_{j=1}^M f_{kj}} \mathbb{E}[Q_m]$.*

Viewing $\mathbb{E}[Q_m]$ as $\xi_m$, the proposition shows that, even if there is no score information available, averaging over query embeddings still offers a way of categorical weighting in terms of label proportions: the weighting coefficients are $\frac{f_{km}}{\sum_{j=1}^M f_{kj}}, m = 1, \ldots, M$. Note that a constraint is the constant $n$ over categories. The constraint can be relaxed if we know the number of samples from each category. Furthermore, when a pool of pairwise comparison results is gathered, one can build a rank over LLMs to determine the best matching model. Therefore, the $\{\mathcal{G}_k\}$ setting and the proposed model embedding mechanism in this section fit naturally into many industrial applications.

## 5  EXPERIMENTS

We implemented FGTS.CDB shown in Alg. 1. Sampling from the posterior $p^j(\theta \mid S_{t-1})$ in Step 5 is implemented by Stochastic Gradient Langevin Dynamics (Welling & Teh

---

[5]Sorting $\{s_{1,m}, s_{2,m}, \ldots, s_{K,m}\}$ results in $s_{(1),m} \geq s_{(2),m} \geq \ldots \geq s_{(K),m}$, and $s_{(\tau),m}$ will be located at the $\tau$-th position.

[6]Since different settings and datasets are discussed in this paper, the word "label" refers to various types of information associated with a question. It may represent the question's category (the category label), the performance or correctness of each LLM (the performance or correctness label), the pairwise comparison result (the pairwise comparison label), or the model that best responds to the question (the label in this context).

[7]This is a reasonable assumption, for instance, when preferences are determined by user traits.

(2011); SGLD). Four text embedding models are chosen to generate embeddings for queries. They are: OpenAI's `text-embedding-3-large` (OpenAI, 2024), `all-MiniLM-L6-v2` (Sentence-Transformers, 2021a), `all-mpnet-base-v2` (Sentence-Transformers, 2021b), and `intfloat/e5-base` (Wang et al., 2022). In this paper, we refer to the models as `OpenAItext`, `MiniLM`, `mpnet`, and `e5b`, respectively. `OpenAItext` serves as a strong and general-purpose embedding model. We compare its embeddings with those generated by the fine-tuned `MiniLM`, `mpnet`, and `e5b` models to examine their impact on the feature function $\phi(x, a)$ derived from a text model. The fine-tuning for an open-sourced text embedding model is implemented by contrastive learning (Khosla et al., 2020; Reimers & Gurevych, 2019). We first build similar and dissimilar query pairs according to their source category or benchmark. Then, the cosine-similarity loss is used to fine-tune the model. For instance, `eb5_E4` means that `eb5` is fine-tuned for four epochs, and `mpnet_E2` represents `mpnet` fine-tuned for two epochs. [8]

## 5.1 ROUTERBENCH

RouterBench (Hu et al., 2024) is a comprehensive benchmark designed for evaluating LLM routing methods. It provides over 405k precomputed inference outputs from eleven diverse LLMs across seven tasks (MMLU, MT-Bench, MBPP, HellaSwag, Winogrande, GSM8K, ARC). The dataset includes detailed performance and cost metadata, enabling systematic analysis of routing strategies. The metadata, organized into a table in Hu et al. (2024) is included as Tab. 3 in App. B.1.

In the following, we describe how to use the metadata and the queries in each benchmark to construct an LLM embedding $a_k$. The learning process is divided into two phases: an offline fine-tuning phase, during which the model embeddings are learned, and an online testing phase, where a realization of FGTS.CDB is evaluated through a sequence of queries. To apply the proposed method, we need to compute $s_k$ and $\xi$. Each benchmark $m$ is treated as a distinct category. To compute the corresponding category embedding $\xi_m$, we apply a text embedding model to five queries sampled from benchmark $m$, and then take the average of their embeddings. These sampled queries are excluded from the online learning phase to prevent data leakage. Suppose `e5b_E4` is applied to generate the embeddings. Then, we obtain $\xi^{(e5b\_E4)} = \left( \xi_{\text{MMLU}}^{(e5b\_E4)}, \xi_{\text{MT-Bench}}^{(e5b\_E4)}, \cdots, \xi_{\text{ARC}}^{(e5b\_E4)} \right)$.

Table 1: Scores of `Perf_cost`, `Excel_perf_cost`, and `Excel_mask`

| LLM | MMLU | | | MT-Bench | | | MBPP | | | HellaSwag | | | Winogrande | | | GSM8k | | | ARC | | |
|---|---|---|---|---|---|---|---|---|---|---|---|---|---|---|---|---|---|---|---|---|---|
| | (i) | (ii) | (iii) | (i) | (ii) | (iii) | (i) | (ii) | (iii) | (i) | (ii) | (iii) | (i) | (ii) | (iii) | (i) | (ii) | (iii) | (i) | (ii) | (iii) |
| WizardLM 13B | 0.562 | 0 | 0 | 0.796 | 0 | 0 | 0.363 | 0 | 0 | 0.600 | 0 | 0 | 0.510 | 0 | 0 | 0.492 | 0 | 0 | 0.657 | 0 | 0 |
| Mistral 7B | 0.558 | 0 | 0 | 0.779 | 0 | 0 | 0.349 | 0 | 0 | 0.517 | 0 | 0 | 0.561 | 0 | 0 | 0.399 | 0 | 0 | 0.640 | 0 | 0 |
| Mixtral 8x7B | 0.721 | 0.721 | 1 | 0.920 | 0.920 | 1 | 0.572 | 0.572 | 1 | 0.634 | 0 | 0 | 0.673 | 0.673 | 1 | 0.485 | 0 | 0 | 0.837 | 0.837 | 1 |
| Code Llama 34B | 0.553 | 0 | 0 | 0.795 | 0 | 0 | 0.464 | 0 | 0 | 0.431 | 0 | 0 | 0.612 | 0 | 0 | 0.424 | 0 | 0 | 0.635 | 0 | 0 |
| Yi 34B | 0.727 | 0.727 | 1 | 0.937 | 0.937 | 1 | 0.331 | 0 | 0 | 0.834 | 0.834 | 1 | 0.743 | 0.743 | 1 | 0.509 | 0.509 | 1 | 0.873 | 0.873 | 1 |
| GPT-3.5 | 0.700 | 0.700 | 1 | 0.907 | 0.907 | 1 | 0.649 | 0.649 | 1 | 0.695 | 0.695 | 1 | 0.623 | 0.623 | 1 | 0.543 | 0.543 | 1 | 0.844 | 0.844 | 1 |
| Claude Instant V1 | 0.368 | 0 | 0 | 0.862 | 0 | 0 | 0.547 | 0 | 0 | 0.704 | 0.704 | 1 | 0.507 | 0 | 0 | 0.561 | 0.561 | 1 | 0.812 | 0 | 0 |
| Llama 70B | 0.629 | 0 | 0 | 0.853 | 0 | 0 | 0.300 | 0 | 0 | 0.627 | 0 | 0 | 0.498 | 0 | 0 | 0.486 | 0 | 0 | 0.784 | 0 | 0 |
| Claude V1 | 0.312 | 0 | 0 | 0.920 | 0.920 | 1 | 0.497 | 0 | 0 | -0.131 | 0 | 0 | 0.516 | 0 | 0 | 0.099 | 0 | 0 | 0.798 | 0 | 0 |
| Claude V2 | 0.456 | 0 | 0 | 0.840 | 0 | 0 | 0.567 | 0.567 | 1 | -0.554 | 0 | 0 | 0.392 | 0 | 0 | -0.011 | 0 | 0 | 0.454 | 0 | 0 |

We then use the metadata in Tab. 3 and the induced Tab. 1 to calculate the scores $s_k$. The most straightforward way is to take only the performance columns in Tab. 3, which corresponds to `Perf`. For instance, taking the performance values in the fifth row of Tab. 3, we obtain $s_{\text{Yi 34B}}^{(\text{Perf})} = (0.743, 0.938, \ldots, 0.882)$. We also implement three other scoring ways, `Perf_cost`, `Excel_perf_cost`, and `Excel_mask`. Using the Perf and the Cost columns in Tab. 3, scores of `Perf_cost` is calculated by $\text{Perf} - \lambda\text{Cost}$ with $\lambda = 0.05$ being a balance parameter. The resulting values are listed in the columns of Tab. 1 indexed by (i). With `Perf_cost` in hand, `Excel_perf_cost` keeps the original `Perf_cost` score if it is ranked as the top-$\tau$ in the column and assigns 0 otherwise. Here we choose $\tau = 3$. The `Excel_perf_cost` scores are listed under index (ii). `Excel_mask` further masks the nonzero values of `Excel_perf_cost`

---

[8]Note that we do not assume access to per-model scalar performance labels during training of the router. Prior high-accuracy LLM routers we are aware of, including universal and cross-attention routers, CARROT, GraphRouter, and EmbedLLM (Jitkrittum et al., 2025; Pulishetty et al., 2025; Somerstep et al., 2025; Feng et al., 2025b; Zhuang et al., 2025), require such labels to train a classifier or regressor. In our preference-only setting these labels are intentionally unavailable, so these methods cannot be applied directly as baselines. Instead, we treat OpenAI's `text-embedding-3-large` as a strong representation-learning baseline and compare it against CCFT-based representations within the same FGTS.CDB framework.

as 1 and lists them under (iii) in Tab. 1. Finally, we feed the scores $s_k$ and the model embeddings $\xi$ to (3), (4), and (5) to obtain the model embedding $a_k$ Specifically, `Perf` and `Perf_cost` are fed to (3), `Excel_perf_cost` to (4), and `Excel_mask` to (5). Using the above symbols, we can denote, for instance, the model embedding of Yi 34B calculated via text embedding model `e5b_E4`, scoring method `Perf_cost`, and weighting equation (3) as $a_{\text{Yi 34B}}^{(\text{e5b\_E4\_Perf\_cost})} = \xi^{(\text{e5b\_E4})} \text{softmax}\left(s_{\text{Yi 34B}}^{(\text{Perf\_cost})}\right)$ [9]. In addition, we append all 14 metadata (Perf and Cost over seven benchmarks) of an LLM at the end of its embedding. Finally, the feature function $\phi(x, a_k)$ is computed as the normalized Hadamard product $x * a_k$. To compare the effectiveness of fine-tuning, we use a suffix to indicate whether a fine-tuned model or an original model generates an embedding. For instance, the string "`e5b_E4_Excel_perf_cost_exp`" means "belonging to the experimental group, each embedding is generated by `e5b` fine-tuned with four epochs with the `Excel_perf_cost` weighting mechanism", and "`e5b_E4_Excel_perf_cost_ctrl`" means "belonging to the control group, each embedding is generated by the original `e5b` with the `Excel_perf_cost` weighting mechanism. Since we cannot fine-tune `OpenAItext`, we use handcraft prompts to generate LLM embeddings. We plug the offline query samples and the meta-data of the dataset into the prompt (Listing 3 in App. D) to get the model description, and then feed it to `OpenAItext` to obtain a model embedding. We use performance metadata as the utility function, from which we generate online feedback via the BLT protocol and compute regret at each round.

We run `OpenAItext` with one, three, and five queries, each for five runs, and average the cumulative regrets to plot three curves [10] in Fig. 2a. For open-sourced embdeeing models, we fine-tune to obtain `e5b_E2`, `e5b_E4`, `mpnet_E2`, `mpnet_E4`, `MiniLM_E2`, and `MiniLM_E4`. For each embedding model, four weighting mechanisms (`Perf`, `Perf_cost`, `Excel_perf_cost`, `Excel_mask`) are implemented. The corresponding control group also undergoes the online testing phase. There are a total of 8 curves for one embedding model. The best performed `e5b_E4` is shown in Fig. 2b. Results for all models can be found in Fig. 6, App. B.2.

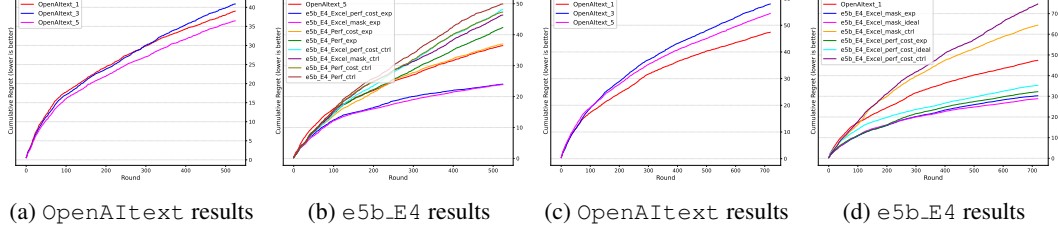

(a) `OpenAItext` results     (b) `e5b_E4` results     (c) `OpenAItext` results     (d) `e5b_E4` results

Figure 2: Regret curves for RouterBench (a, b) and robust generalization (c, d).

We obtain the following observations from the regret curves. In Fig. 2a, the number of queries in the prompt does not affect `OpenAItext` too much. Thus, we choose the best version, `OpenAItext_5`, to compare with the results generated by our proposed CCFT in Fig. 2b. First, the experimental group outperforms the control group, showing the advantage of fine-tuning. Second, `e5b_E4_Excel_perf_cost_exp` and `e5b_E4_Excel_mask_exp` outperform `OpenAItext_5`, showing that even an open-sourced model, with a careful design, can generate better embeddings than a strong general-purpose model. Third, when examining all plots in Fig. 6b – 6f, the previous two observations hold for all other embedding models, showing convincing evidence for the effectiveness of CCFT we proposed. Fourth, by comparing `e5b_E4_Excel_perf_cost_exp` with `e5b_E4_Perf_cost_exp`, we observe that weighting via `Excel_perf_cost` yields better performance than weighting via `Perf_cost`. This suggests that, instead of weighting over all categories, it is more effective to weight only the categories in which the LLM demonstrates expertise. In addition to achieving lower accumulated regret, both `Excel_perf_cost` and `Excel_mask` incorporate a performance-cost trade-off, enhancing their practical applicability and flexibility through

---

[9]Note that `e5b`, `mpnet`, and `MiniLM` are text models that generate embeddings. They need to be distinguished from the LLMs, listed in the leftmost column of Tab. 1, which are candidates selected to generate responses in online learning.

[10]Unless mentioned, each regret curve reported is the average of 5 runs.

the tunable parameter $\lambda$. App. B.3 presents a comparison between our approach and the closely related MixLLM (Wang et al., 2025).

### 5.1.1 GENERALIZATION TO AN UNSEEN BENCHMARK

Although an online learning setting evaluates an algorithm's adaptivity by providing sequentially and randomly shuffled inputs, the algorithm may still access metadata from all benchmarks during the offline stage. To more rigorously assess adaptivity to an unseen benchmark, we modify the data-generation pipeline to ensure that one benchmark remains completely hidden from the algorithm throughout both the offline and online phases. The queries and metadata of MT-Bench are removed, as its dataset is not large enough to induce a distribution shift scenario in the coming experiment. For the remaining six benchmarks, the metadata for ARC is removed from Tab. 3, ensuring that the algorithm is oblivious to it from the outset. An online learning sequence is composed of two sections. First, we sample 60 queries from each benchmark, excluding ARC (i.e., five benchmarks in total), resulting in 300 queries. These are randomly shuffled to form the first section of the sequence. Next, for the second section, we sample 120 queries from ARC and an additional 300 non-overlapping queries from the other five benchmarks, following the same procedure as in the first section. These 420 (i.e., $120 + 300$) queries are then shuffled to form the second section of the learning sequence. This setup introduces a shift in the query distribution during the second section of online learning, as queries from a previously unseen benchmark are added to evaluate the robust generalization capability of the proposed method. Due to the modification in how metadata is accessed, we introduce an additional suffix, `ideal`, which allows the model to access ARC's metadata. Although the `ideal` case is not realistic in practice, comparing results from configurations ending with `ideal` and those ending with `exp` enables us to assess the adaptivity strength of our method. For the rest of the experimental setting, we follow the same fine-tuning protocol in the last section to generate embeddings. Based on the observations from figures 2b and 6, we implement the `Excel_perf_cost` and `Excel_mask` weighting schemes due to their consistently strong performance.

The results of `OpenAItext` and `e5b` are shown in figures 2c and 2d. Results for `mpnet` and `MiniLM` can be found in Fig. 7 from App. B.2. We choose `OpenAItext_1` to compare with the results generated by `e5b_E4` according to Fig. 2c. First, obviously, the fine-tuning group outperforms the control group. Second, `e5b_E4_Excel_perf_cost_exp` and `e5b_E4_Excel_mask_exp`, the fine-tuning results via our CCFT strategy outperform that of `OpenAItext_1`. The case in the second observation holds for Fig. 7b through Fig. 7f, justifying the effectiveness of the proposed method. Third, interestingly, we find that it is not always the case that an `ideal` curve outperforms the corresponding `exp` curve. This can be found by comparing the pair (`e5b_E4_Excel_perf_cost_exp`, `e5b_E4_Excel_perf_cost_ideal`) with the pair (`e5b_E4_Excel_mask_exp`, `e5b_E4_Excel_mask_ideal`). The situation also can be found in Fig. 7. This phenomenon suggests weighting less may be better than weighting more, and it may be related to the last observation we have in the original RouterBench experiments that weighting over all benchmark embeddings is not a good idea. Maybe the metadata alone is not enough to make a weighting judgment; we might need to look into other aspects of the benchmarks in future work.

### 5.2 MIXINSTRUCT

MixInstruct (Jiang et al., 2023) is a 110K-example instruction-following benchmark built to evaluate LLM routing methods. It mixes data from four sources (Alpaca-GPT4, Dolly-15K, GPT4All-LAION, ShareGPT) with a 100k/5k/5k train/dev/test split. The authors run eleven popular open-source LLMs on the full set, then derive oracle pairwise preferences by prompting ChatGPT to compare every candidate pair per example.

Table 2: Models Ranked First by Percentage of Examples

| Model | Vicuna | MOSS | Open Assistant | Alpaca | Baize | ChatGLM | MPT | Koala | Dolly V2 | StableLM | FLAN-T5 |
|---|---|---|---|---|---|---|---|---|---|---|---|
| Percentage (%) | 21.22 | 12.91 | 12.61 | 11.61 | 11.61 | 8.51 | 7.61 | 6.71 | 4.50 | 1.90 | 0.80 |

Tab. 2, adapted from Figure 1 of Jiang et al. (2023), underscores the importance of LLM routing: selecting the best-matching LLM for each query is crucial, as any fixed-LLM strategy will yield no more than 22% accuracy. A distinctive characteristic of the MixInstruct dataset is the absence of an explicit category label, rendering the proposed methods in (3), (4), and (5) infeasible. Since

this challenge can naturally arise in practical settings, we use MixInstruct to evaluate our alternative formulation presented in (6), thereby validating its applicability under such a realistic constraint. Moreover, the pairwise comparison labels in MixInstruct make it closely resemble datasets commonly encountered in industrial applications.

To make the regret defined in (1) computable, we reconstruct the utility function $r^*(x, a)$. Pairwise comparisons between LLM candidates are translated into scores by adding a win value of 1, a tie 0.5, and a loss 0. For each query we then take the LLM with the highest score as its "label" in the sense of § 4.2; queries sharing the same top-scoring LLM are grouped into $\mathcal{G}_k$, and the model embeddings $a_k$ are computed via (6). During the translation process, a Condorcet winner may emerge (Black, 1958; Wikipedia contributors, 2025). To ensure it receives the highest score, we assign the Condorcet winner a top score with an additional bonus. During the data analysis stage, we found the existence of ambiguous queries. We applied the OpenAI API to assign an ambiguity score to each query and removed the most ambiguous 8% and 15% of queries. The ambiguity removal process introduces additional sub-strings _8 and _15 in the naming of regret curves. Using (6), the rest of the experiment setup follows the procedure described in § 5.1. We obtain Fig. 3 and deferred all results to App. C.

In Fig. 3a, we again observe that implementations of the proposed (6) outperform OpenAItext variants, demonstrating that our approach remains effective even when the dataset lacks metadata information. When comparing results under different degrees of ambiguity removal, we observe a consistent pattern in Fig. 3b across e5b_E4, mpnet_E4, and OpenAItext_5: removing the top

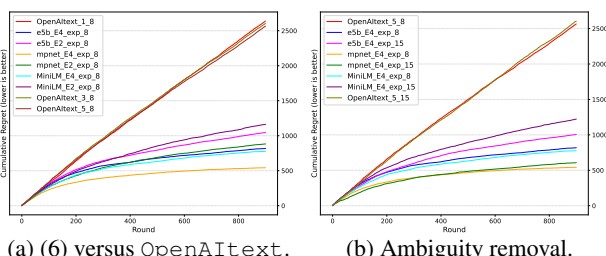

(a) (6) versus OpenAItext.  (b) Ambiguity removal.

Figure 3: Regret curves for MixInstruct.

15% of ambiguous queries is worse than removing only the top 8%. This highlights the risk of discarding learnable information when too many queries are removed.

## 6 CONCLUDING REMARKS AND FUTURE WORK

We proposed CCFT, an embedding learning strategy that aligns prospective queries with model expertise through category-calibrated representations. Four variants of CCFT were implemented and combined with the theoretically grounded FGTS.CDB algorithm to form the first trainable contextual dueling learner for LLM routing. The proposed methods were systematically evaluated on two real-world datasets, RouterBench and MixInstruct, demonstrating their effectiveness. Our approach also exhibits robust generalization and achieves a performance-cost balance, both of which are critical for practical deployment.

Looking forward, we highlight three promising directions for future work. First, as noted in § 5.1.1, our current model representation is effective but may not fully capture the potential of LLM expertise. Enhancing this alignment between query semantics and model capabilities could lead to even better routing performance. We plan to further pursue this direction by exploring new factors and techniques for representing model expertise. Second, although our method is designed for pairwise feedback, we conjecture that it can be adapted to work with pointwise feedback as well. However, building a unified system that can effectively integrate both types of supervision remains an open challenge. Addressing this would offer both practical value and deeper academic understanding. Finally, while our work is based on Feel-Good Thompson Sampling for contextual dueling bandits (Li et al., 2024) (because of the strong empirical performance shown in Li et al. (2024) over other approaches), our approach could be combined with alternative algorithms such as UCB-style contextual dueling bandits, and systematically comparing these variants is also an interesting direction for future work.

**Ethics statement**  This work represents an original contribution derived from our research on LLM routing. We have made an honest and balanced effort to report both the strengths and limitations of the proposed methods. The study is primarily methodological and theoretical in nature. The text embedding models used are either open-source academic resources or publicly available commercial products. All datasets employed are publicly accessible and widely accepted within the research community. Therefore, we do not anticipate any immediate risks of misuse or harm to human society arising from this work.

**Reproducibility statement**  We list below the datasets and text embedding models used in the experiments reported in this submission.

- MMLU https://openreview.net/forum?id=d7KBjmI3GmQ
- RouterBench https://openreview.net/forum?id=IVXmV8Uxwh
- MixInstruct https://aclanthology.org/2023.acl-long.792/
- text-embedding-3-large of OpenAI https://platform.openai.com/docs/guides/embeddings
- all-MiniLM-L6-v2 of Sentence-Transformers https://huggingface.co/sentence-transformers/all-MiniLM-L6-v2
- all-mpnet-base-v2 of Sentence-Transformers https://huggingface.co/sentence-transformers/all-mpnet-base-v2
- intfloat/e5-base https://arxiv.org/abs/2212.03533

We will release the code necessary to reproduce the experiments, including data processing, embedding learning, and algorithm evaluation. To ensure reproducibility, we have seeded all random processes using the following code block.

```python
def set_seed(seed):
    random.seed(seed)
    np.random.seed(seed)
    torch.manual_seed(seed)
    if torch.cuda.is_available():
        torch.cuda.manual_seed_all(seed)
        torch.backends.cudnn.deterministic = True
```

Listing 1: A Python function to ensure reproducibility.

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

## A SUPPLEMENTARY MATERIALS FOR SECTION 4

### A.1 EXPERIMENTAL DETAILS OF MMLU

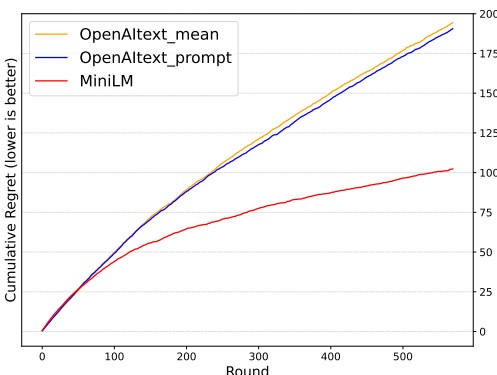

Figure 4: Two failed examples (`OpenAItext_mean` and `OpenAItext_prompt`) versus a successful example (`MiniLM`).

This section explains how the curves in Fig. 4 are constructed. We chose five topics, abstract algebra, anatomy, astronomy, international law, and machine learning, from MMLU (Hendrycks et al., 2021). Queries are sampled to form two disjoint offline learning and online testing sets. For each topic, ten queries are sampled for offline learning. The online samples for each topic are drawn in proportion to the dataset, forming an online test set of 595 queries in total.

**The Construction of `OpenAItext_prompt` and `OpenAItext_mean`** Since MMLU does not involve LLMs, we need to construct our own LLM experts and the corresponding performance values. A straightforward way is to assume there are five LLMs, each with expertise in one of the topics. Using offline queries, we explore two approaches to generate the model embeddings $a_k$[11]. In the first way, we encode the model description via OpenAI's `text-embedding-3-large`, where the model description is the combination of our handcrafted prompt with offline queries (Listing 2 in App. D). We use `OpenAItext_prompt` to denote its results. In the second way, we first generate offline query embeddings of a topic using `text-embedding-3-large` and then take their average as the model embedding. `OpenAItext_mean` is used to denote its results. The intuition behind both approaches is to represent a model using the sample queries it excels at. It is rather simple to generate a query embedding $x$, as we simply feed the query string into `text-embedding-3-large`. We tested the Hadamard product (element-wise multiplication) $\phi(x, a) = x * a$ with normalization and vector addition (element-wise addition) $\phi(x, a) = x + a$ with normalization to construct $\phi$ and keep the first one based on the experimental outcomes. To construct performance values, we compute a similarity matrix using the average query embeddings for each topic and the cosine similarity function. Given the topic of the current query and the algorithm's selections, we can retrieve the corresponding similarity scores from this matrix to quantify the algorithm's performance. These similarity scores are then used both to sample feedback via the BTL model and to generate the performance values needed for computing regret at each round.

**The Construction of `MiniLM`** `MiniLM` is an abbreviation of `all-MiniLM-L6-v2`. Its construction follows the procedure as `OpenAItext_mean`, with two modifications. First, the embedding model is replaced with `all-MiniLM-L6-v2`. Second, contrastive learning (Khosla et al., 2020; Reimers & Gurevych, 2019) is applied to fine-tune the model. To do so, we construct similar and dissimilar query pairs based on their source category, and fine-tune the model using a cosine similarity loss for four epochs. The regret curve corresponding to this model is labeled as `MiniLM`.

We note that Fig. 1 is identical to Fig. 4, and that `MiniLM` is omitted from the main text. The goal there is to illustrate how a successful routing method should behave, rather than to define the

---

[11]For consistency, we use $a_k$ both to index an LLM (as in § 3) and to denote its embedding (as explained next), with the intended meaning clear from context.

method itself. Since `MiniLM` is not the final version of our proposed approach, and MMLU is not an ideal benchmark for evaluating routing strategies, we chose to defer these details to the appendix. Nonetheless, MMLU is sufficiently simple to serve as a synthetic simulation for demonstrating our motivation.

## A.2 PROOF OF PROPOSITION 1

*Proof of Proposition 1.* Let $k$ be fixed. Assume without loss of generality that each $f_{km}n$ is an integer, the size of $\mathcal{G}_k$ is $\sum_{j=1}^{M} f_{kj}n$. Then,

$$\frac{\sum_{q \in \mathcal{G}_k} q}{|\mathcal{G}_k|} = \frac{\sum_{m=1}^{M} \sum_{q \in \mathcal{G}_k \cap \mathcal{Q}_m} q}{\sum_{j=1}^{M} f_{kj}n} = \sum_{m=1}^{M} \frac{f_{km}n}{\sum_{j=1}^{M} f_{kj}n} \left( \frac{\sum_{q \in \mathcal{G}_k \cap \mathcal{Q}_m} q}{f_{km}n} \right).$$

The term in the parentheses is the sample average of $f_{km}n$ independent embeddings drawn from category $m$. Hence, it is an unbiased estimator of $\mathbb{E}[Q_m]$ and we have the proposition. $\square$

**Example (Interpretation of Prop. 1).** To illustrate Prop. 1, consider $M = 2$ latent categories, each containing $n$ queries. Suppose that for a fixed LLM $k$ we have $f_{k1} = 0.75$ and $f_{k2} = 0.25$, i.e., model $k$ is the best-matching LLM for $75\%$ of the queries in category 1 and for $25\%$ of the queries in category 2. Then $|\mathcal{G}_k| = (0.75 + 0.25)n$ and Eq. 6 averages the embeddings of these $(0.75 + 0.25)n$ queries. Prop. 1 tells us that, in expectation as $n$ grows, this average converges to

$$a_k \approx \frac{0.75}{0.75 + 0.25} \mathbb{E}[Q_1] + \frac{0.25}{0.75 + 0.25} \mathbb{E}[Q_2],$$

that is, a convex combination of the category-level mean embeddings weighted by how often model $k$ is preferred in each category. This is precisely the kind of categorical reweighting we aimed to achieve, but obtained here without explicitly observing category labels or performance scores.

## A.3 T-SNE VISUALIZATIONS

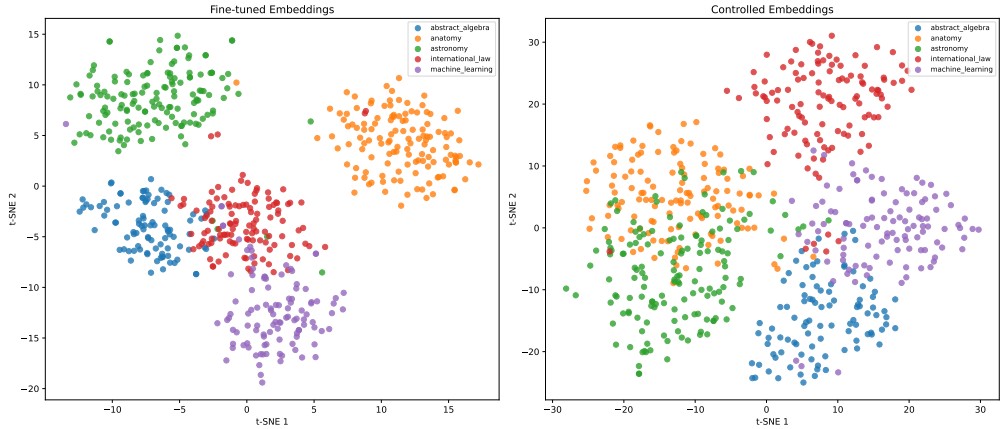

Figure 5: 2D t-SNE visualization of the embeddings generated by the fine-tuned MiniLM model discussed in Appendix A.1 (left), compared to those without fine-tuning (right). Each point represents an embedding projected into 3D space, with colors indicating cluster membership.

# B ROUTERBENCH SUPPLEMENTARY MATERIALS

## B.1 TABLE 1 OF HU ET AL. (2024)

Tab. 3 is identical to Table 1 in Hu et al. (2024). We include it here for completeness.

Table 3: Performance and cost across benchmarks

| LLM | MMLU | | MT-Bench | | MBPP | | HellaSwag | | Winogrande | | GSM8k | | ARC | |
|---|---|---|---|---|---|---|---|---|---|---|---|---|---|---|
| | Perf↑ | Cost↓ | Perf↑ | Cost↓ | Perf↑ | Cost↓ | Perf↑ | Cost↓ | Perf↑ | Cost↓ | Perf↑ | Cost↓ | Perf↑ | Cost↓ |
| WizardLM 13B | 0.568 | 0.122 | 0.796 | 0.006 | 0.364 | 0.011 | 0.636 | 0.727 | 0.512 | 0.040 | 0.510 | 0.354 | 0.660 | 0.068 |
| Mistral 7B | 0.562 | 0.081 | 0.779 | 0.003 | 0.349 | 0.006 | 0.541 | 0.485 | 0.562 | 0.027 | 0.409 | 0.210 | 0.642 | 0.046 |
| Mixtral 8x7B | 0.733 | 0.245 | 0.921 | 0.012 | 0.573 | 0.023 | 0.707 | 1.455 | 0.677 | 0.081 | 0.515 | 0.594 | 0.844 | 0.137 |
| Code Llama 34B | 0.569 | 0.317 | 0.796 | 0.015 | 0.465 | 0.021 | 0.525 | 1.882 | 0.617 | 0.104 | 0.462 | 0.752 | 0.644 | 0.177 |
| Yi 34B | 0.743 | 0.326 | 0.938 | 0.018 | 0.333 | 0.031 | 0.931 | 1.938 | 0.748 | 0.107 | 0.552 | 0.867 | 0.882 | 0.182 |
| GPT-3.5 | 0.720 | 0.408 | 0.908 | 0.026 | 0.651 | 0.044 | 0.816 | 2.426 | 0.630 | 0.134 | 0.601 | 1.170 | 0.855 | 0.228 |
| Claude Instant V1 | 0.384 | 0.327 | 0.863 | 0.030 | 0.550 | 0.064 | 0.801 | 1.943 | 0.512 | 0.108 | 0.626 | 1.300 | 0.821 | 0.183 |
| Llama 70B | 0.647 | 0.367 | 0.854 | 0.022 | 0.302 | 0.039 | 0.736 | 2.183 | 0.504 | 0.121 | 0.529 | 0.870 | 0.794 | 0.205 |
| Claude V1 | 0.475 | 3.269 | 0.938 | 0.361 | 0.527 | 0.607 | 0.841 | 19.43 | 0.570 | 1.077 | 0.653 | 11.09 | 0.889 | 1.829 |
| Claude V2 | 0.619 | 3.270 | 0.854 | 0.277 | 0.605 | 0.770 | 0.421 | 19.50 | 0.446 | 1.081 | 0.664 | 13.49 | 0.546 | 1.833 |
| GPT-4 | 0.828 | 4.086 | 0.971 | 0.721 | 0.682 | 1.235 | 0.923 | 24.29 | 0.858 | 1.346 | 0.654 | 19.08 | 0.921 | 2.286 |

## B.2 Additional Results for RouterBench

Fig. 6 presents the cumulative regret curves for `e5b_E4`, `e5b_E2`, `mpnet_E4`, `mpnet_E2`, `MiniLM_E4`, `MiniLM_E2`, and the `OpenAItext` variants. Fig. 6h compares all embedding models under the `Excel_perf_cost` and `Excel_mask` mechanisms, which represent the most effective weighting methods in most cases. Note that Fig. 6a is identical to Fig. 2b, and Fig. 6g is identical to Fig. 2a. They are included here for completeness.

## B.3 Comparison with MixLLM

We compare our work with the most relevant related method, MixLLM, proposed by Wang et al. (2025). Both approaches adopt online learning frameworks with binary feedback. However, there are three fundamental differences between the two.

First, MixLLM uses pointwise feedback (e.g., like/dislike) as input, whereas our method relies on pairwise feedback (i.e., preference comparisons). As a result, the problem settings are inherently different. Second, MixLLM employs an upper confidence bound (UCB)-based strategy, where uncertainty is managed via the matrix $A_l$ (see (9) in their paper). In contrast, our approach is based on TS, where uncertainty is governed by posterior sampling and the likelihood function $L^j$ (2). Third, our method requires significantly fewer offline training samples. Specifically, we use only five queries per benchmark (thirty-five in total) for offline learning. According to Table 1 in Wang et al. (2025), MixLLM requires at least 30% of the dataset for offline training. This sample efficiency is an appealing feature of our approach.[12]

## B.4 Additional Results for Robust Generalization

Fig. 7 shows how `e5b_E4`, `e5b_E2`, `mpnet_E4`, `mpnet_E2`, `MiniLM_E4`, `MiniLM_E2`, and the `OpenAItext` variants adapt to the unseen ARC benchmark. Fig. 7h collects the regret curves of all models implemented by `Excel_mask` and `Excel_perf_cost` weighting mechanisms. Note that Fig. 7a is identical to Fig. 2d, and Fig. 7g is identical to Fig. 2c. They are included here for completeness.

## B.5 Additional Results of Fine-tuning without Categorical Weighting

Fig. 8 shows the regret curves for `e5b_E4`, `e5b_E2`, `mpnet_E4`, `mpnet_E2`, `MiniLM_E4`, `MiniLM_E2`, and the `OpenAItext` without categorical weighting. To remove categorical weighting, we take `Excel_mask` and set $\tau = 1$.

## C MixInstruct Supplementary Materials

## C.1 Additional Results

Figure 9a, which is identical to Figure 3a, compares all text models with the top 8% most ambiguous queries removed. Figure 9b presents results with the top 15% of ambiguous queries removed. Fig-

---

[12]In particular, we use five queries per category in the current section, fifteen for robust generalization evaluation (§ 5.1.1), and ten queries for the MixInstruct experiments (§ 5.2).

ure 9c, identical to Figure 3b, compares the effects of removing different proportions of ambiguous queries.

## D  PROMPTS FOR OPENAI'S TEXT-EMBEDDING-3-LARGE MODEL

The prompt in Listing 2 is used to generate model embeddings in § 4.1. The prompt in Listing 3 is used to generate model embeddings in § 5.

```python
prompt = (
    f"This model is very good at solving questions regarding {category}."
    f"Example questions it excels at: "
    f"1. {example_questions[0]}"
    f"2. {example_questions[1]}."
)
```

Listing 2: The Python code block including the prompt used in MMLU.

```python
avg_perf = np.mean(aggregated_data[model_benchmark]["Perf"])
avg_cost = np.mean(aggregated_data[model_benchmark]["Cost"])
cost_efficiency = 1 / avg_cost if avg_cost > 0 else float("inf")

qs = example_qs[:return_id+1]

if len(qs) > 1:
    questions = ", ".join(qs[:-1]) + f", and {qs[-1]}"
else:
    questions = qs[0]

prompt = (
    f"This is {model_name}, a language model with "
    f"average performance score of {avg_perf:.3f} "
    f"and cost efficiency rating of {cost_efficiency:.3f}."
    f"It has shown particular strength in {model_benchmark} type
    ↪  questions."
    f"Example question(s) it handles: {questions}."
)
```

Listing 3: The Python code block including the prompts used in RouterBench and MixInstruct.

## E  THE USE OF LARGE LANGUAGE MODELS

We used ChatGPT-4o and ChatGPT-5 to assist with the following tasks:

- Writing support, including wording suggestions, sentence smoothing, and grammar checking
- Table generation
- Figure arrangement and layout improvement, including tips for enhancing visualization
- Literature review during the initial and drafting stages of the project

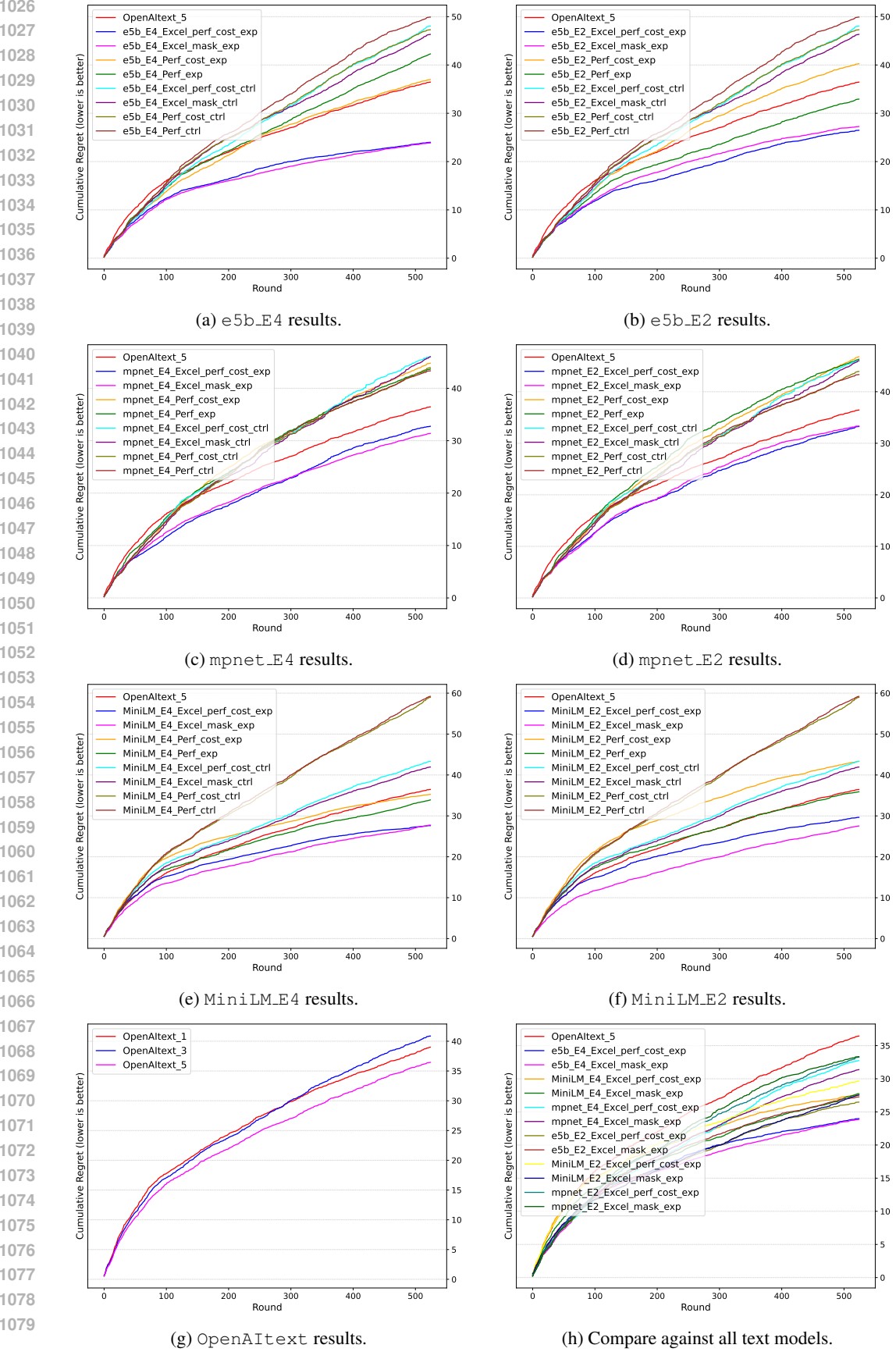

(a) e5b_E4 results.

(b) e5b_E2 results.

(c) mpnet_E4 results.

(d) mpnet_E2 results.

(e) MiniLM_E4 results.

(f) MiniLM_E2 results.

(g) OpenAItext results.

(h) Compare against all text models.

Figure 6: Cumulative regret curves for RouterBench.

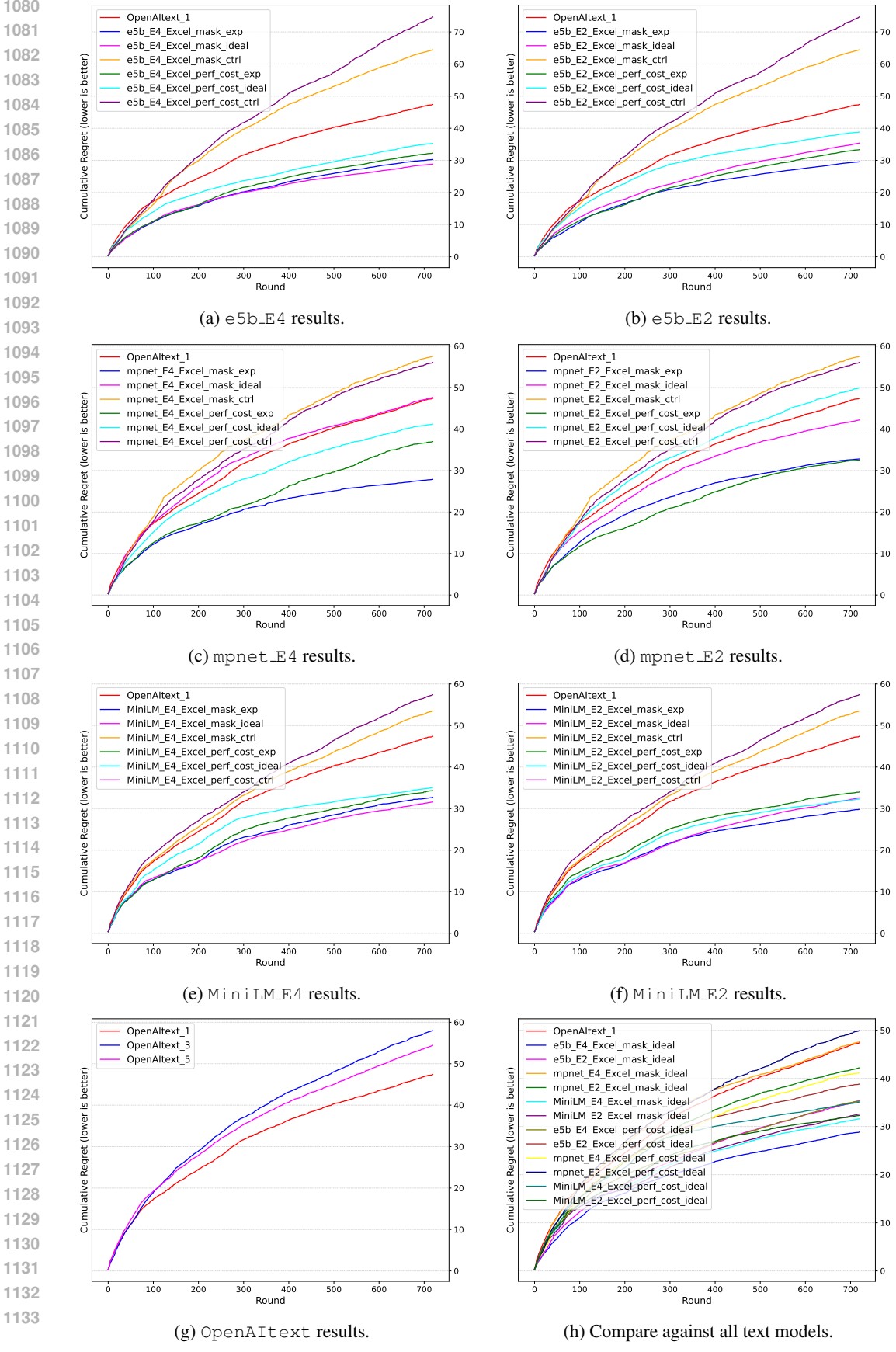

Figure 7: Cumulative regret curves for robust generalization.

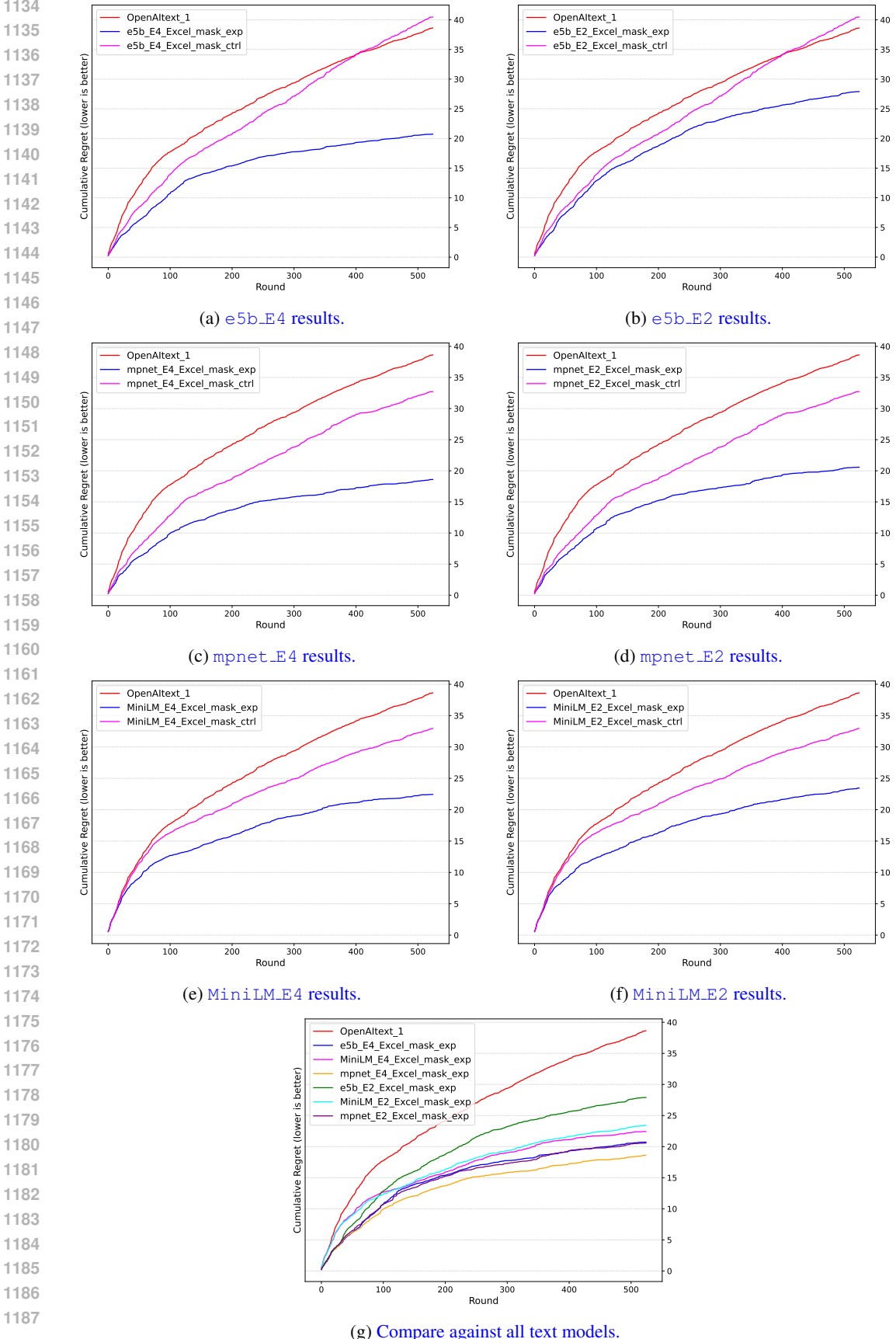

(a) e5b_E4 results.

(b) e5b_E2 results.

(c) mpnet_E4 results.

(d) mpnet_E2 results.

(e) MiniLM_E4 results.

(f) MiniLM_E2 results.

(g) Compare against all text models.

Figure 8: Cumulative regret curves for fine-tuning without categorical weighting.

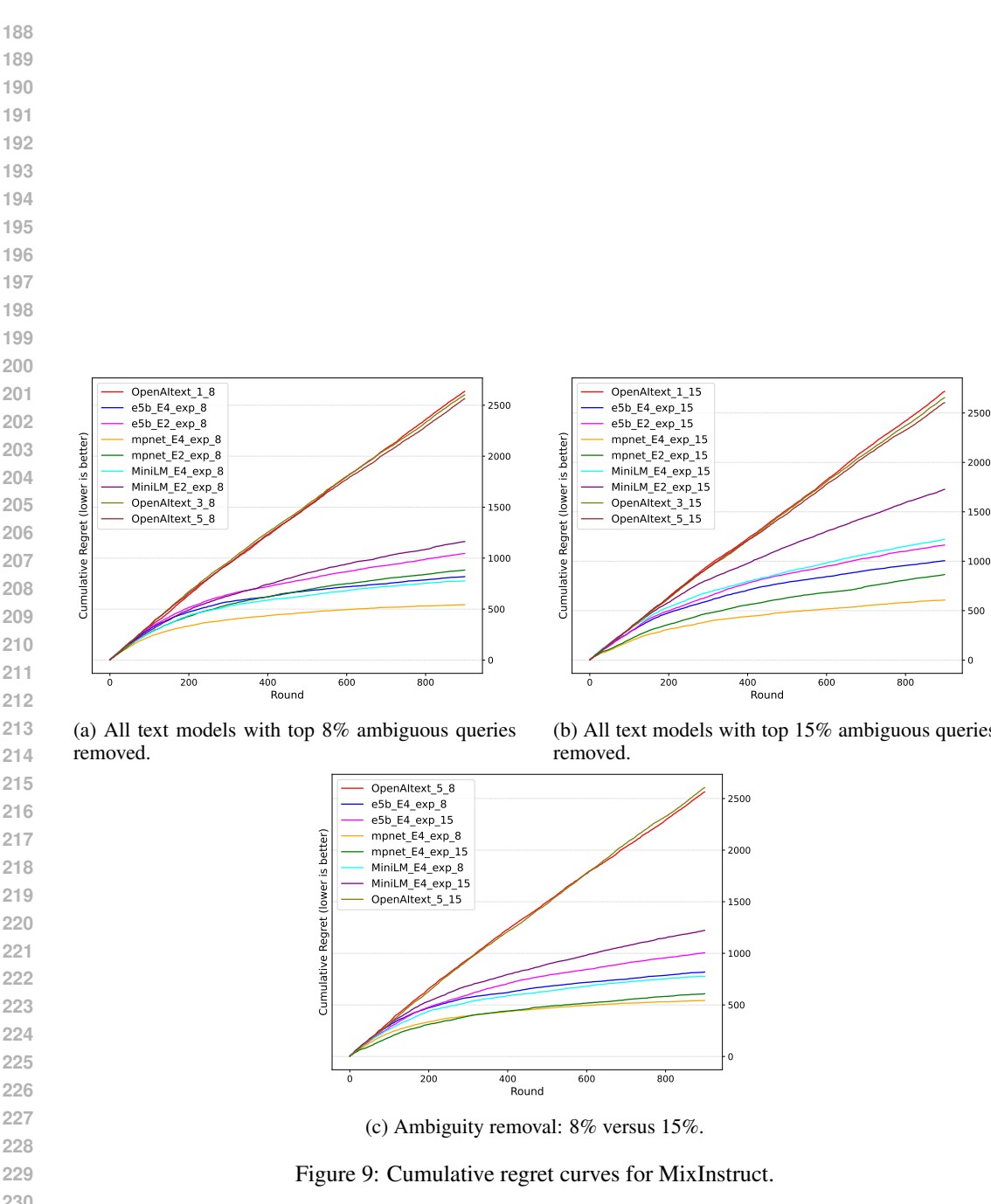

(a) All text models with top 8% ambiguous queries removed.

(b) All text models with top 15% ambiguous queries removed.

(c) Ambiguity removal: 8% versus 15%.

Figure 9: Cumulative regret curves for MixInstruct.

