# OpenReview forum: "LLM Routing with Dueling Feedback"
_ICLR.cc/2026/Conference — Submitted to ICLR 2026_

### Official Review · Reviewer_iqVs · 2025-10-18

**Soundness:** 4
**Presentation:** 3
**Contribution:** 3
**Rating:** 6
**Confidence:** 4

**Summary:**

This paper tackles the LLM routing problem that is dynamically selecting the best-fit large language model for each query under constraints of user satisfaction, model expertise, and cost. The authors cast routing as a contextual dueling bandit task, where the system learns from pairwise preference feedback instead of absolute quality scores. They introduce a novel Category-Calibrated Fine-Tuning (CCFT) method to learn model-representative embeddings via contrastive fine-tuning with categorical weighting, encoding each model’s domain expertise. These embeddings enable an implementation of Feel-Good Thompson Sampling for Contextual Dueling Bandits (FGTS.CDB), a Thompson-sampling algorithm adapted to preference feedback. The approach is evaluated on two benchmarks, RouterBench (a multi-task LLM routing dataset) and MixInstruct (an instruction-following benchmark). The experiments show that the proposed routing strategy achieves lower cumulative regret and faster convergence than baselines using generic embeddings, while maintaining better robustness and a favorable performance-vs-cost tradeoff. In summary, the paper’s contributions include formulating LLM routing as a dueling bandit, a new CCFT embedding strategy (with four variants integrating model quality and cost), and a theoretically grounded FGTS-based router that outperforms strong baselines on diverse tasks.

**Strengths:**

- Problem Formulation: The paper tackles LLM routing in a novel way by combining online adaptive learning with preference-based weak supervision. This joint treatment of adaptivity and pairwise feedback fills an important gap, as previous work had not simultaneously addressed both aspects. The use of dueling bandits (pairwise comparisons) for model selection is a creative and timely idea, reflecting how user feedback can be more practically obtained as comparisons rather than absolute ratings.

- Theoretical Foundation: The approach is built on a solid theoretical framework. It leverages Feel-Good Thompson Sampling (FGTS), which has known robust guarantees for bandits, and extends it to contextual dueling settings. The paper provides theoretical justification (Proposition 1) linking the learned embeddings to unbiased utility estimates, and clearly explains how binary preference feedback maps onto an underlying utility function of user satisfaction, expertise, and cost.

- Effective Methodology (CCFT): The proposed Category-Calibrated Fine-Tuning strategy is a general yet powerful way to encode model specialization. By fine-tuning language model embeddings on representative queries per category, the router’s context features become highly informative of which model will perform well. This is the first demonstration of a trainable contextual dueling bandit for LLM routing. Technically, the paper integrates techniques like contrastive learning and softmax-weighted embeddings in an elegant manner, and the algorithmic solution (FGTS.CDB) is well-motivated.

- Empirical Results: The experiments are thorough and convincing. On two benchmarks (RouterBench and MixInstruct), the authors show that their approach consistently achieves lower cumulative regret and faster convergence than baselines, meaning it learns the optimal routing policy more efficiently. The methods are compared across multiple embedding models (open-source vs OpenAI) and across different weighting schemes, demonstrating the approach’s robustness.

- Practical Efficiency: The method is sample-efficient and adaptable. It requires only a small amount of offline data for fine-tuning – e.g., using merely 5 example queries per category to calibrate the embeddings, which is far less than what some prior approaches needed. This low data requirement makes the approach practical to deploy.

**Weaknesses:**

- Lack of Real-World User Validation: A limitation is that the feedback mechanism is only evaluated in simulation or with automated oracles, rather than with human users. For RouterBench, the “preferences” are presumably derived from known performance metrics, and for MixInstruct the pairwise labels are generated by prompting an LLM as an oracle. While this is a reasonable proxy, it remains uncertain how the routing algorithm would perform with actual user feedback, which can be noisy or inconsistent. In practice, obtaining pairwise comparisons may require showing multiple model outputs to users or running interleaved A/B tests, which the paper does not explore. Demonstrating the approach in a real user study or on an online platform would strengthen the claims of practical utility.

- Complexity and Weighting Scheme Efficacy: The proposed solution introduces several moving parts including fine-tuning embeddings, computing category-weighted representations with various “Perf cost” or “Excel” weighting schemes, and then running a specialized bandit algorithm. This complexity could hinder real-world deployment and is not thoroughly ablated for necessity. In particular, the categorical weighting variants sometimes show mixed results.

- Baseline Comparisons: While the paper compares to a strong baseline (using a general-purpose OpenAI embedding in the same FGTS bandit framework) and outperforms it, the scope of baselines is somewhat limited. There is no direct comparison to a conventional supervised routing approach or to other online bandit algorithms in the literature (e.g., UCB-based methods or the MixLLM approach). In fact, the authors only discuss MixLLM qualitatively in the appendix, noting differences in feedback type and data requirements, but do not provide a head-to-head empirical comparison (likely because the problem settings differ). Including or adapting a few alternative routing strategies (such as a static classifier trained on a portion of data, or a simpler bandit without the special embedding) would bolster the empirical evaluation.

- Focus on Pairwise Feedback Only: The solution is tailored to preference feedback, which is a reasonable design given the motivation, but it does not incorporate pointwise feedback (individual ratings or correctness signals) or hybrid approaches. In real applications, one might have access to occasional explicit ratings or other supervision in addition to pairwise comparisons. The authors acknowledge in discussion that FGTS.CDB is designed for pairwise data and conjecture it could be extended to pointwise signals, but they do not implement or test such extensions. This is a missed opportunity to demonstrate flexibility.

**Questions:**

- Collecting Pairwise Feedback in Practice: How do you envision obtaining the pairwise preference feedback from real users in an online setting? The paper argues that binary comparisons are more user-friendly than absolute ratings, but implementing this in a one-shot routing scenario is non-trivial as users typically only see one model’s answer. Would the system occasionally present two answers for the same query and ask the user to choose (akin to an A/B test)? If so, how would you manage the added latency and cost?

- “Excel” Weighting Strategy Insight: The results indicate that the more complex weighting schemes (e.g., Excel perf cost) did not universally outperform simpler ones, and in some cases weighting fewer top categories yielded better outcomes. Could you elaborate on why that might be? For instance, does the “Excel” thresholding effectively filter out noise, or could it be discarding useful signal from less prominent categories? What does this imply about the quality of the metadata (performance/cost) being used?

- Integration of Pointwise Feedback: Do you have plans to extend or adapt your approach to also utilize pointwise feedback signals? Since you conjecture that FGTS.CDB could be modified for pointwise rewards, it would be interesting to hear how this might be achieved.

---

> ### Author Response · Authors · 2025-11-21
> **Response to Reviewer iqVs: Part 1**
>
> We greatly appreciate your thoughtful and constructive comments. We have carefully addressed each concern, and our detailed responses are presented below.
>
> _____
>
> **W1.** Lack of Real-World User Validation: A limitation is that the feedback mechanism is only evaluated in simulation or with automated oracles, rather than with human users. For RouterBench, the “preferences” are presumably derived from known performance metrics, and for MixInstruct the pairwise labels are generated by prompting an LLM as an oracle. While this is a reasonable proxy, it remains uncertain how the routing algorithm would perform with actual user feedback, which can be noisy or inconsistent. In practice, obtaining pairwise comparisons may require showing multiple model outputs to users or running interleaved A/B tests, which the paper does not explore. Demonstrating the approach in a real user study or on an online platform would strengthen the claims of practical utility.
>
> **R1.** This is true. However, as an academic study, especially one of the initial papers in the binary feedback direction, we choose to focus on proof of concept. Thus, we formulate the problem in a restricted way compared to your suggestion. We evaluate our method on two well-known datasets to test its applicability and reproducibility. When a practical dataset comes out, we are of course glad to put our method to test. We will discuss this as one of our future work.
>
> _____
>
> **W2.** Complexity and Weighting Scheme Efficacy: The proposed solution introduces several moving parts including fine-tuning embeddings, computing category-weighted representations with various “Perf cost” or “Excel” weighting schemes, and then running a specialized bandit algorithm. This complexity could hinder real-world deployment and is not thoroughly ablated for necessity. In particular, the categorical weighting variants sometimes show mixed results.
>
> **R2.** We are not proposing a single method to solve all routing problems. We want to demonstrate the potential of the contextual dueling approach in solving routing problems with an extremely low amount of information in the feedback. The construction of CCFT shows its flexibility and effectiveness. A practitioner might need to try different configurations to find the suitable embedding for the problem at hand.
>
> _____
>
> **W3.** Baseline Comparisons: While the paper compares to a strong baseline (using a general-purpose OpenAI embedding in the same FGTS bandit framework) and outperforms it, the scope of baselines is somewhat limited. There is no direct comparison to a conventional supervised routing approach or to other online bandit algorithms in the literature (e.g., UCB-based methods or the MixLLM approach). In fact, the authors only discuss MixLLM qualitatively in the appendix, noting differences in feedback type and data requirements, but do not provide a head-to-head empirical comparison (likely because the problem settings differ). Including or adapting a few alternative routing strategies (such as a static classifier trained on a portion of data, or a simpler bandit without the special embedding) would bolster the empirical evaluation.
>
> **R3.** For baselines and MixLLM comparisons, please refer to the general reply (A). A short reply is that for a baseline that requires the performance information for learning, it is unfair to test it under our setting since we ask it to learn beyond its limit (incomparable settings).
>
> Practically, a TS-based method is preferred over a UCB-based method. Since we have already discovered a working method, it is unnecessary to propose another one. There is no obvious way to adapt MixLLM to our problem setting, because the feedback in MixLLM is “like or dislike,” whereas ours is a preference between two options. Indeed, a UCB-based method could be a candidate, but exploring the potential of UCB-based methods is a separate study from the current one.
>
> _____
>
> **W4.** Focus on Pairwise Feedback Only: The solution is tailored to preference feedback, which is a reasonable design given the motivation, but it does not incorporate pointwise feedback (individual ratings or correctness signals) or hybrid approaches. In real applications, one might have access to occasional explicit ratings or other supervision in addition to pairwise comparisons. The authors acknowledge in discussion that FGTS.CDB is designed for pairwise data and conjecture it could be extended to pointwise signals, but they do not implement or test such extensions. This is a missed opportunity to demonstrate flexibility.
>
> **R4.** We believe contributing a solution for the binary preference feedback setting is sufficient, as no previous work has explored this direction. Extending to a hybrid (pointwise and preference) feedback setting is an interesting future direction.
>
> _____

---

> > ### Author Response · Authors · 2025-11-21
> > **Response to Reviewer iqVs: Part 2**
> >
> > **Q1.** Collecting Pairwise Feedback in Practice: How do you envision obtaining the pairwise preference feedback from real users in an online setting? The paper argues that binary comparisons are more user-friendly than absolute ratings, but implementing this in a one-shot routing scenario is non-trivial as users typically only see one model’s answer. Would the system occasionally present two answers for the same query and ask the user to choose (akin to an A/B test)? If so, how would you manage the added latency and cost?
> >
> > **R1.** Answer 1: I would search literature for a problem setting that an agent needs to keep making decisions but with some kind of delayed feedback.
> >
> > Answer 2: I would assume the user base is large enough; otherwise, data insufficiency is unavoidable. Then I can assign different users to different A/B test instances. If the users share similar preferences, the collected data will be close to the setting in this paper.
> >
> > _____
> >
> > **Q2.** “Excel” Weighting Strategy Insight: The results indicate that the more complex weighting schemes (e.g., Excel perf cost) did not universally outperform simpler ones, and in some cases weighting fewer top categories yielded better outcomes. Could you elaborate on why that might be? For instance, does the “Excel” thresholding effectively filter out noise, or could it be discarding useful signal from less prominent categories? What does this imply about the quality of the metadata (performance/cost) being used?
> >
> > **R2.** From the algorithm’s design idea, the success comes from aligning LLM expertise with question type. Imagine a case: LLM$_1$ is good at type-A questions and LLM$_2$ is good at type-B. In such a case, Excel would outperform Excel_perf_cost, as there is no reason to assign type-A questions to LLM$_2$, and vice versa. Conversely, if different LLMs exhibit varying performance across individual questions within the same question type, then Excel_perf_cost, which incorporates performance metadata, becomes more meaningful. To validate these observations, it is necessary to construct different controlled situations (datasets). This would be a valuable direction for follow-up investigation.
> >
> > _____
> >
> > **Q3.** Integration of Pointwise Feedback: Do you have plans to extend or adapt your approach to also utilize pointwise feedback signals? Since you conjecture that FGTS.CDB could be modified for pointwise rewards, it would be interesting to hear how this might be achieved.
> >
> > **R3.** This is a conjecture: pointwise feedback could be viewed as pairwise feedback with some fixed reference LLM.
> >
> > _____

---

### Official Review · Reviewer_hEWF · 2025-10-27

**Soundness:** 3
**Presentation:** 2
**Contribution:** 3
**Rating:** 4
**Confidence:** 3

**Summary:**

This paper studies LLM routing problem and formulate this problem as contextual dueling bandit learning from pairwise preference feedback. To leverage dueling bandit algorithm, this work also introduces category-calibrated fine-tuning to derive model embedding. Empirical results are presented with four different types of the category weighting with the state-of-the-art feel-good Thompson sampling for contextual dueling bandit algorithm. Across benchmarks, the proposed methods can achieve lower cumulative regret over generic embedding models.

**Strengths:**

1. This paper studies an interesting and practical problem: how to route user queries to the best LLM. I found this problem has great potential and influence in the future. Also it is interesting to formulate this problem based on dueling bandit to leverage sequential learning and pairwise preference.
2. It is new for me to use dueling bandit for this problem setting.
3. The empirical results are complete and prove that the proposed methods can beat baselines.

I am not familiar with this line of LLM routing works so I may refer to other reviewers' opinions and the rebuttal to adjust my rating accordingly.

**Weaknesses:**

1. I feel the presentation can still be improved, especially Section 4.2. It is still not very clear to me how did your fine-tuning work with the model embedding $a_k$ you deduced there. It seems that you have already deduced the model embedding formulation, and then how did you fine-tune the small embedding models with contrastive learning? The training datasets are not very clear to me. The last paragraph of page 5 (line 261-266) is also not very clear to readers, and it is a bit hard to understand.

2. After reading some related works before reviewing this work, it seems that there is a line of methods considering model/LLM embedding along with the query embedding for the LLM routing problem ([R1, R2]). And I feel the methodology is kinda similar with this work: cluster the user queries and take the (weighted) average performance for each LLM. It is better to highlight the differences here.

3. It might be better to include some modern baselines (e.g. Universal routing, CARROT) to showcase the superiority of the proposed methods. Although most of them consider offline training framework instead of online learning, you can still compare your method with those baselines in a testing datasets after convergence.

R1: Universal Model Routing for Efficient LLM Inference, Jitkrittum et al.
R2: One Head, Many Models: Cross-Attention Routing for Cost-Aware LLM Selection, Pulishetty et al.

**Questions:**

Why did you choose to use Thompson sampling for the bandit algorithm part? I think UCB is also a popular and state-of-the-art contextual bandit framework.

---

> ### Author Response · Authors · 2025-11-21
> **Response to Reviewer hEWF**
>
> We greatly appreciate your thoughtful and constructive comments. We have carefully addressed each concern, and our detailed responses are presented below.
>
> ___
>
> **W1.** I feel the presentation can still be improved, especially Section 4.2. It is still not very clear to me how did your fine-tuning work with the model embedding $a_k$ you deduced there. It seems that you have already deduced the model embedding formulation, and then how did you fine-tune the small embedding models with contrastive learning? The training datasets are not very clear to me. The last paragraph of page 5 (line 261-266) is also not very clear to readers, and it is a bit hard to understand.
>
> **R1.** Thank you for the suggestions. Please refer to the general replies (B) and (C). (B) explains the fine-tuning of CCFT, and (C) explains the fine-tuning in lines 261 to 266.
>
> _____
>
> **W2.** After reading some related works before reviewing this work, it seems that there is a line of methods considering model/LLM embedding along with the query embedding for the LLM routing problem ([R1, R2]). And I feel the methodology is kinda similar with this work: cluster the user queries and take the (weighted) average performance for each LLM. It is better to highlight the differences here.
>
> **R2.**
> UniRoute [R1] took a cluster based approach to compute the performance of an LLM. However, their method relies on the loss \ell(x, y, \hat{y}) that cannot be computed since the optimal response is not available in our setting (the only learning signal in our setting is the binary preference feedback).
>
> In [R2], the score used to train the attention-based predictor is the ground-truth performance score. This real value score is also unavailable in our setting.
>
> _____
>
> **W3.** It might be better to include some modern baselines (e.g. Universal routing, CARROT) to showcase the superiority of the proposed methods. Although most of them consider offline training framework instead of online learning, you can still compare your method with those baselines in a testing datasets after convergence.
>
> **R3.** We received similar questions from other reviewers, and have prepared a general reply (A). A brief answer is that it is unknown how to train the UniRoute [R1] and CARROT [R3] given the binary preference feedback.
>
> In CARROT [R3], the weight vector $\mu$ is similar to our perf_cost but can take more metrics. However, the metrics predictor $\Phi$ requires $Y$ for regression. As long as $Y$ cannot be recovered by the binary preference feedback, $\Phi$ is hard to learn. Currently, given only binary preference feedback, it is unclear how to construct $Y$.
>
> _____
>
> ### References
>
> [R1] Jitkrittum et al. (2025). Universal Model Routing for Efficient LLM Inference.
> https://arxiv.org/pdf/2502.08773
>
> [R2] Pulishetty et al. (2025). One Head, Many Models: Cross-Attention Routing for Cost-Aware LLM Selection.
> https://arxiv.org/pdf/2509.09782
>
> [R3] Somerstep et al. (2025). CARROT: A Cost Aware Rate Optimal Router.
> https://arxiv.org/pdf/2502.03261
>
> _____
>
>
> **Q1.** Why did you choose to use Thompson sampling for the bandit algorithm part? I think UCB is also a popular and state-of-the-art contextual bandit framework.
>
> **R1.** Practically, a Thompson sampling (TS)-based method is preferred over a UCB-based method. Since this paper mainly considers real-world datasets, we prioritize TS over UCB. Indeed, a UCB-based method could be a candidate, but exploring the potential of UCB-based methods is a separate study from the current paper, as we have already discovered a working method.
> As an example where FGTS performs better than others including UCB-based methods, please see Table 1 and Figure 1 of Li et al. "Feel-Good Thompson Sampling for Contextual Dueling Bandits" (ICML 2024) https://openreview.net/pdf?id=l9ga3iQuHt However, we agree that it would be interesting to explore these alternative directions in the future, and we have added this as future work in the final section of our paper.

---

> > ### Comment · Reviewer_hEWF · 2025-11-25
> >
> > Thank you for your response. Could you also reply to my question in the "Questions" section: Why did you choose to use Thompson sampling for the bandit algorithm part? I think UCB is also a popular and state-of-the-art contextual bandit framework.
> >
> > Besides, it would be better if the authors could upload the revised manuscript to fix the presentation problem, and I believe ICLR allows one more page for the rebuttal revision. Right now, the main paper still seems very unclear to me.

---

> ### Author Response · Authors · 2025-11-25
> **Thank you for the comment**
>
> Dear reviewer hEWF,
>
> Thank you for reading our replies, and apologies that we didn't include the question in our original reply. We have updated **Response to Reviewer hEWF** to address your question on UCB. Please refer to **R1** of **Q1**. For reference we copy the same reply below:
>
> > R1. Practically, a Thompson sampling (TS)-based method is preferred over a UCB-based method. Since this paper mainly considers real-world datasets, we prioritize TS over UCB. Indeed, a UCB-based method could be a candidate, but exploring the potential of UCB-based methods is a separate study from the current paper, as we have already discovered a working method. As an example where FGTS performs better than others including UCB-based methods, please see Table 1 and Figure 1 of Li et al. "Feel-Good Thompson Sampling for Contextual Dueling Bandits" (ICML 2024) https://openreview.net/pdf?id=l9ga3iQuHt However, we agree that it would be interesting to explore these alternative directions in the future, and we have added this as future work in the final section of our paper.
>
> Furthermore, we are grateful to have the chance to revise the manuscript with the additional page for rebuttal revision. We hope to share our updated paper in 1 or 2 days.
>
> Best,
>
> Authors

---

> > ### Author Response · Authors · 2025-11-26
> >
> > Dear reviewer hEWF,
> >
> > We would like to let you know that we have revised the manuscript in light of your comments and suggestions. The changes are shown in blue text color. We appreciate your helpful suggestion and the opportunity to revise the manuscript, and we hope that the changes address your concerns.
> >
> > Best,
> >
> > Authors

---

> > > ### Comment · Reviewer_hEWF · 2025-11-26
> > >
> > > Thank you for your response. I encourage the authors to polish the Section 4.2 as the original illustration is not very clear. I have no more question and would raise my rating.

---

### Official Review · Reviewer_RdwD · 2025-10-30

**Soundness:** 2
**Presentation:** 3
**Contribution:** 3
**Rating:** 6
**Confidence:** 3

**Summary:**

The paper proposes CCFT for encoding LLM expertise and applies FGTS.CDB, based on CCFT-derived LLM embeddings, for online LLM routing under pairwise preference feedback.

**Strengths:**

1.	This work first studied LLM routing under online pairwise preference feedback, which meets the needs of practical applications.
2.	The proposed method has been empirically proven to improve cumulative regret.
3.	The writing is basically clear.

**Weaknesses:**

1.	Since prior works (Feng et al. 2025, Zhuang et al. 2025) have proposed techniques to strengthen users' and models' semantic information, it would be better to discuss why these previous approaches cannot be directly applied or what advantages the proposed CCFT offers.
2.	The experiments can be further strengthened. Apart from cumulative regret, the paper should include more intuitive evaluation metrics (e.g, accuracy on different tasks) to demonstrate the effectiveness of the proposed method. The work can consider two more baselines for ablation study, namely, OpenAItext with categorical weighting, e5b fine-tuning but without categorical weighting to construct model embedding. The standard derivation is better to report for understanding the model performance stability.


Minor
1.	Not clear how to compute regret. Line 795-798 is hard to follow.
2.	It is hard to tell which one is better in Figure 5. It may provide 2D t-SNE.
3.	In A.2, $m$ should be $M$.
4.	Line 256 should be three variants.

**Questions:**

1.	How is preference feedback collected in the experiments? Lines 353 and 433 mention it, but the process is still unclear.
2.	Confusion about the usage of pairwise comparison labels mentioned in Line 446-447.
3.	Confusion about how to calculate model embedding for the MixInstruct experiment when there is no explicit category label. From my understanding, Eq. (6) is still based on the label information as mentioned in Line 263-264.
4.	What distribution is considered for $p_0$ in Algorithm 1?
5.	How many queries pairs required for fine-tuning text embedding models?

---

> ### Author Response · Authors · 2025-11-21
> **Response to Reviewer RdwD: Part 1**
>
> We greatly appreciate your thoughtful and constructive comments. We have carefully addressed each concern, and our detailed responses are presented below.
>
> _____
>
> **W1.** Since prior works (Feng et al. 2025, Zhuang et al. 2025) have proposed techniques to strengthen users' and models' semantic information, it would be better to discuss why these previous approaches cannot be directly applied or what advantages the proposed CCFT offers.
>
> **R1.** Thank you for the suggestion. We argue that both prior works cannot be applied directly to the problem studied in this paper.
> 1. There is no clear way to calculate the performance column required in Figure 1 of Feng et al. (2025) under our setting, where only binary preference feedback is provided.
> 2. Zhuang et al. (2025) require the correctness matrix Y to learn the embedding functions, but it is unclear how to convert a binary preference feedback to a correctness label.
>
> _____
>
> **W2.** The experiments can be further strengthened. Apart from cumulative regret, the paper should include more intuitive evaluation metrics (e.g, accuracy on different tasks) to demonstrate the effectiveness of the proposed method. The work can consider two more baselines for ablation study, namely, OpenAItext with categorical weighting, e5b fine-tuning but without categorical weighting to construct model embedding. The standard derivation is better to report for understanding the model performance stability.
>
> **R2.**
> 1. Regarding reporting accuracy, please refer to the general reply (A).
> 2. For the ablation study, we are running experiments on fine-tuning without categorical weighting using e5b, mpnet, and MiniLM. The results can be found in Appendix B.5 and Figure 8. Since RouterBench does not contain the metadata of OpenAItext, we can’t apply categorical weighting for it.
> 3. Usually, standard deviation is not plotted for a multi-armed bandit method since the regret is cumulative and scales with rounds T, so does the deviation, which will lead to misinterpretation.
>
> _____
>
> **Minor 1.** Not clear how to compute regret. Line 795-798 is hard to follow.
>
> **R1.** Since MMLU does not involve LLM, we let an LLM be a topic and define $r^{\*}(x, a)$ as the similarity between the topic of $x$ and the predicted topic $a$. Thus, looking at (1) (with subscript $t$ removed), $r^{\*}(x, a^{\*})$ is always 1 since the optimal prediction $a^{\*}$ is the topic of the question $x$, and $r^{\*}(x, a^1)$ (resp. $r^{\*}(x, a^2)$) is the similarity between the topic of $x$ and topic $a^1$ (resp. the similarity between the topic of $x$ and topic $a^2$). For five topics, a 5$\times$5 cosine similarity matrix is sufficient and can speed up the computation. Lastly, we average question embeddings of the same topic to represent the topic.
> - Thank you for the opportunity to clarify this point, and we will update the explanation in lines 795 to 798.
>
> _____
>
> **Minor 2.** It is hard to tell which one is better in Figure 5. It may provide 2D t-SNE.
>
> **R2.** We have replaced Figure 5 with a 2D t-SNE. Thank you.
>
> _____
>
> **Minor 3.** In A.2, $m$ should be $M$.
>
> **R3.** Thank you for pointing this out. We have made the correction.
>
> _____
>
> **Minor 4.** Line 256 should be three variants.
>
> **R4.** Four variants are: (i) perf and (ii) perf_cost generated from (3), (iii) excel_perf_cost from (4), and (iv) excel_mask from (5).
>
> _____

---

> ### Author Response · Authors · 2025-11-21
> **Response to Reviewer RdwD: Part 2**
>
> **Q1.** How is preference feedback collected in the experiments? Lines 353 and 433 mention it, but the process is still unclear.
>
> **R1.**
> 1. When we choose LLM$_i$ and LLM$_j$, we can look up their performance values from the metadata (Table 3 in Appendix B.1 with the row indexed by LLM and the column indexed by category). These values correspond to the utility values $r^{\*}(x, a^i)$ and $r{\*}(x, a^j)$ that are used to compute $\mathbb{P}(y=1|x, a^i, a^j)$ in BTL (line 157).
> 2. In line 433, we recall how the preference feedback is generated in MixInstruct: the authors take a pair of responses and ask ChatGPT to provide the preference.
>
> We will revise the main text to enhance clarity.
>
> _____
>
> **Q2.** Confusion about the usage of pairwise comparison labels mentioned in Line 446-447.
>
> **R2.** We have the ideal case in MixInstruct, where all 55 comparisons are available (MixInstruct includes 11 models, hence totally 11*10/2 = 55 pairs of responses). In a real-world application, we can collect many (query, LLM$_i$, LLM$_j$, preference feedback) tuples and reorganize the data to form a data set similar to MixInstruct. Therefore, our proposed method may have its application usage.
>
> _____
>
> **Q3.** Confusion about how to calculate model embedding for the MixInstruct experiment when there is no explicit category label. From my understanding, Eq. (6) is still based on the label information as mentioned in Line 263-264.
>
> **R3.** Please refer to the general reply (C). The short answer is that we use different terms, such as high-quality label and categorical label, to denote different types of annotation in different settings. We will make a note in the main text to avoid confusion.
>
> _____
>
> **Q4.** What distribution is considered for $p_o$ in Algorithm 1?
>
> **R4.** We consider the Gaussian distribution. If another distribution is used, only the likelihood function needs to be modified; the rest of the algorithmic framework remains unchanged.
>
> _____
>
> **Q5.** How many queries pairs required for fine-tuning text embedding models?
>
> **R5.** Take the RouterBench experiment as an example. For each category (benchmark), we sample 5 questions. Therefore, there are 5\*4/2 = 10 similar pairs for each category. For negative pairs, we sample at most 7\*10 = 70 pairs to ensure data balance. In total, we have 10\*7+70 = 140 pairs for fine-tuning. Note that 7 is the number of benchmarks in RouterBench.
> - Please refer to the last paragraph of the general reply (B) for numbers used in the other experiments. We apply the data-balancing mechanism in all our experiments.
>
> _____

---

> > ### Comment · Reviewer_RdwD · 2025-11-26
> >
> > Thanks for the detailed responses.
> >
> > I still feel that relying solely on cumulative regret is insufficient to evaluate the efficacy of the proposed method, especially given the extensive literature on LLM routing. At a minimum, the authors should report accuracy (or other relevant metrics) for the proposed pipeline and use the strong LLM’s performance on the same evaluation set (or random selection as in [1]) as a reference point. In addition, the other works [1, 2, 3] cited in this paper also utilize preference data; some of their techniques may be adapted as baselines for the setting studied here, thus also addressing concerns from other reviewers regarding the need for comparisons with more modern baselines.
> >
> > [1] Isaac Ong, Amjad Almahairi, Vincent Wu, Wei-Lin Chiang, Tianhao Wu, Joseph E. Gonzalez, M. Waleed Kadous, and Ion Stoica. RouteLLM: Learning to Route LLMs from Preference Data. In International Conference on Learning Representations, pp. 1–14, 2025.
> > [2] Zesen Zhao, Shuowei Jin, and Zhuoqing Morley Mao. Eagle: Efficient Training-Free Router for
> > Multi-LLM Inference. In Proceedings of the Workshop on Machine Learning for Systems at
> > NeurIPS 2024, pp. 1–8, 2024.
> > [3] Xinyuan Wang, Yanchi Liu, Wei Cheng, Xujiang Zhao, Zhengzhang Chen, Wenchao Yu, Yanjie Fu,
> > and Haifeng Chen. MixLLM: Dynamic Routing in Mixed Large Language Models. In Proceed-
> > ings of the 2025 Conference of the Nations of the Americas Chapter of the Association for Compu-
> > tational Linguistics: Human Language Technologies (Volume 1: Long Papers), pp. 10912–10922,
> > 2025.

---

> ### Author Response · Authors · 2025-11-27
>
> Dear Reviewer RdwD,
>
> Thank you for raising these points, and for the helpful suggestions. We are currently deriving the accuracy metrics, and hope to add it before the discussion period ends.
>
> We ran RouterBench experiments with the suggested random selection strategy. We also compare with an optimal selection strategy (that selects two identical LLMs that are optimal every time). The results below show that our e5b_E4 and mpnet_E4 performs close to the optimal with respect to the score, and is much better than the random strategy.
>
> | Method   | Mean over 3 shuffles | Std              |
> | -------- | -------------------- | ---------------- |
> | optimal  | 0.7663               | 0.0085           |
> | e5b_E4   | 0.7536               | 0.0276           |
> | mpnet_E4 | 0.7416               | 0.0291           |
> | random   | 0.5855               | 0.0190           |
>
> Eagle [2], by design (Section 2.2, page 3 in [2]), ranks LLMs only based on empirical scores and does not include any exploration mechanism. As a result, it cannot manage the exploration–exploitation trade-off and is vulnerable to distribution shift in online settings, an issue that our Thompson-sampling-based method is explicitly designed to handle. Because Eagle’s code is not publicly available, conducting full experimental comparisons is difficult. However, in our simple implementation, the cumulative-regret curve grows roughly linearly, consistent with our conjecture regarding its limited ability to adapt online data. If accepted, we will complete the implementation and add the comparison to the camera ready version.
>
> RouteLLM [1] and MixLLM [3] operate in different regimes: [1] uses preference labels similar to our case but handles only 2‑model routing with an offline setup, and [3] utilizes point‑wise supervision. In contrast, we study online routing over >2 models from pairwise feedback. We will clarify these differences in the paper.

---

### Official Review · Reviewer_v6zs · 2025-11-01

**Soundness:** 3
**Presentation:** 2
**Contribution:** 3
**Rating:** 4
**Confidence:** 3

**Summary:**

The authors propose FGTS.CDB to address the LLM routing task. They leverage the Contextual Dueling Bandit paradigm, employing methods from online learning, reinforcement learning and weak supervision to tackle the high costs associated with both data labeling and model inference. Through experiments on the RouterBench and MixInstruct datasets, the proposed model demonstrates lower cumulative regret, faster convergence, and robust generalization, outperforming OpenAI Text embedding model.

**Strengths:**

1. The method utilizes online learning to dynamically update the agent, enabling an adaptive balance in selecting among multiple models, which is a significant advantage over traditional static approaches.
2. By relying on comparative judgments (i.e., pairwise preference feedback), the framework reduces the high cost of annotation and mitigates the problem of inconsistent evaluation criteria that often plagues absolute scoring.
3. Through the use of context-based categorization and weighting (the CCFT method), the approach effectively refines the feature space, leading to more targeted and efficient model selections.

**Weaknesses:**

1. The online learning phase introduces significant cost and latency overhead, which seems to contradict the primary motivation of LLM routing—cost reduction. The dueling method mandates two LLM calls for each learning step, whereas a well-tuned cascading system might succeed with just a single call in most cases.
2. The assumption that pairwise preference feedback can be synthetically generated via a BTL model is overly idealistic. Real-world human feedback is notoriously stochastic, non-stationary, and can exhibit non-transitivity (e.g., a user might prefer A>B and B>C, but also C>A), which violates the BTL model's assumptions. Consequently, the smooth convergence of the regret curves may be an artifact of this idealized simulation.
3. The CCFT method primarily relies on a predefined set of categories. When dealing with cross-domain or novel query types not seen during the offline phase, this rigid structure could lead to suboptimal routing decisions, as the a priori categorization may not generalize well and could mask essential information.
4. The red line in Fig. 1, which is meant to represent a successful learning curve, is not identified in the legend, hindering clarity.

**Questions:**

See weaknesses.

---

> ### Author Response · Authors · 2025-11-21
> **Response to Reviewer v6zs**
>
> We greatly appreciate your thoughtful and constructive comments. We have carefully addressed each concern, and our detailed responses are presented below.
>
> _____
>
> **W1.** The online learning phase introduces significant cost and latency overhead, which seems to contradict the primary motivation of LLM routing—cost reduction. The dueling method mandates two LLM calls for each learning step, whereas a well-tuned cascading system might succeed with just a single call in most cases.
>
> **R1.** There can be many problem formulations for the routing problem. In our setting, the algorithm is *formulated* to output two LLMs but the same LLM can be chosen twice which means we naturally only require a single call in such cases. We see that the regret curve becomes flat as rounds increase, which means the algorithm chooses the same optimal LLM more often. This can be seen in Eq. (1). Otherwise, if one of the LLMs is not optimal, the regret curve will keep climbing. This is why we choose to show the learning curve instead of reporting a final score.
>
> _____
>
> **W2.** The assumption that pairwise preference feedback can be synthetically generated via a BTL model is overly idealistic. Real-world human feedback is notoriously stochastic, non-stationary, and can exhibit non-transitivity (e.g., a user might prefer A>B and B>C, but also C>A), which violates the BTL model's assumptions. Consequently, the smooth convergence of the regret curves may be an artifact of this idealized simulation.
>
> **R2.** BTL is a fundamental assumption in preference modeling, and it appears in both routing and RLHF. We agree with the existence of the non-transitivity or stochastic situations the reviewer mentioned. In fact, BTL naturally captures the situations the reviewer mentioned. In line 157, we can observe $C>A$ or $A>C$ based on the sampled result from $\mathbb{P}(y|x, A, C)$. To verify the effectiveness of BLT, we evaluate our method on two real-world datasets (RouterBench and MixInstruct). We also conduct an unseen generalization test in Section 5.1.1. Results from these three sets of experiments confirm the effectiveness of the proposed BTL-based method.
>
> _____
>
> **W3.** The CCFT method primarily relies on a predefined set of categories. When dealing with cross-domain or novel query types not seen during the offline phase, this rigid structure could lead to suboptimal routing decisions, as the a priori categorization may not generalize well and could mask essential information.
>
> **R3.** To address the concern, experiments in MixInstruct (Section 5.2) do not require categorical information. For RouterBench, we have unseen generalization experiments that verify the proposed method on an unseen benchmark (with one benchmark removed from offline stage) in Section 5.1.1. Both experiments verify the effectiveness of the proposed method addressing your concern on the availability of the categorical information.
>
> _____
>
> **W4.** The red line in Fig. 1, which is meant to represent a successful learning curve, is not identified in the legend, hindering clarity.
>
> **R4.** Thank you. Figure 4 (line 773) corresponds to Figure 1. We have put back the legend.

---

### Author Response · Authors · 2025-11-21
**General Response and Clarifications: (A) and (B)**

We appreciate the reviewers’ time and insightful comments. Before addressing the individual comments, we provide the following general response that clarifies the motivation behind our approach and explains key methodological choices. These clarifications address the main concerns raised across the reviews and highlight the contributions and strengths of our work.
_____

### **(A) Rationale Behind the Proposed Binary-Preference Routing Method**

All referenced papers in the reviews [1-5] produce models with high accuracy since they have access to high-quality labels such as the performance or correctness of each question. Therefore, their effectiveness is unknown under *binary preference feedback*, because we currently do not know how to train them in this restricted setting. To address this fundamental limitation, our study provides a first feasible direction for building a learnable routing system driven solely by binary preference feedback.

**Remark**: "binary preference feedback" is a concise way of referring to "binary preference feedback from pairwise comparisons." When contrasting with point-wise feedback, we explicitly use the term "pairwise feedback" to avoid ambiguity.

Given the unique position of this work, our contributions are twofold:
- A feasible training method, combining Algorithm 1 (FGTS.CDB) and Section 4.2 (CCFT), that enables routing from binary preferences.
- An empirical demonstration of its effectiveness (Section 5).

Because no prior work directly tackles binary preference feedback for routing, baseline comparisons are not available. A reasonable comparison is to compare our learned representations with those generated by OpenAI’s text-embedding model, whose training does not rely on performance values. At this stage, we prioritize effectiveness over accuracy. Once future models trained directly on binary preference feedback become available, accuracy-based comparisons will become meaningful.

### References
[1] Jitkrittum et al. (2025). Universal Model Routing for Efficient LLM Inference.
https://arxiv.org/pdf/2502.08773

[2] Pulishetty et al. (2025). One Head, Many Models: Cross-Attention Routing for Cost-Aware LLM Selection.
https://arxiv.org/pdf/2509.09782

[3] Somerstep et al. (2025). CARROT: A Cost Aware Rate Optimal Router.
https://arxiv.org/pdf/2502.03261

[4] Feng et al. (2025). GraphRouter: A Graph-based Router for LLM Selections.
https://arxiv.org/pdf/2410.03834

[5] Zhuang et al. (2025). EmbedLLM: Learning Compact Representations of Large Language Models.
https://arxiv.org/pdf/2410.02223
_____

### **(B) Overview of CCFT**
The design of our method is guided by the following hypothesis: an LLM’s expertise is characterized by the categories (or benchmarks) in which it performs well, and the semantic meaning of each category is determined by the questions belonging to it. Under this view, CCFT can be intuitively described as follows: we use question embeddings to construct category embeddings, and then combine these category embeddings with weights derived from category-level performance metadata to obtain the LLM embedding.

The actual implementation proceeds in three steps. First, the contrastive fine-tuning is applied only at the question-embedding level, which makes question embeddings within the same category form a denser cluster and separate more clearly from other categories (lines 225 to 228; lines 802 to 805). Second, after fine-tuning, the category embeddings $\xi$ are constructed by averaging the in-category question embeddings (lines 228 to 231), and no additional fine-tuning is required here. Note that $\xi$ is a vector of category embeddings (line 236). Third, with the category embeddings $\xi$ available, equations (3) to (5) provide three strategies for combining category embeddings into a single LLM embedding using category-level metadata.
- We will make a note in lines 225–228 to refer the reader to the contrastive fine-tuning in lines 802–805.

An advantage of this approach is that we only need a small amount of offline question sets for us to perform contrastive fine-tuning.
- For RouterBench (Section 5.1), the offline set contains 5 sampled questions from each benchmark.
- For unseen generalization verification (Section 5.1.1), the offline set contains 15 sampled questions from each benchmark.
- For MixInstruct (Section 5.2), for each model we sample 10 questions on which it is the top-performing model to form the offline set.

The reason why our approach work is that the linear reward model (line 162) separates the LLM expertise ($\phi(x, a)$) from user preference ($\theta^{\*}$), so that we can use a small offline question set and the metadata of a dataset to learn $\phi(x, a)$ and then learn the $\theta^{\*}$ in the online stage using binary preference feedback (note that in (2) we only need the binary feedback y to update the posterior).
_____

---

> ### Author Response · Authors · 2025-11-21
> **General Response and Clarifications: (C)**
>
> ### **(C) Model Embedding without Categorical Information**
>
> Assume there are M latent categories (unknown to us). Uniformly sampled questions form a union of $Q_1$ to $Q_M$. Although we do not know which question belongs to which category, we do know the best preferred model for each question; we use “label” to refer to the best preferred model of a question (line 261). (In MixInstruct, we have all pairwise comparison results that allow us to rank LLMs for each question.)
> - In the paper, we use different terms such as high-quality label and categorical label to denote different types of annotation in different settings. We will make a note in the main text to avoid confusion.
>
> (lines 261–267) For each LLM$_k$, we gather the questions in the union of $Q_1$ to $Q_M$ for which it ranks top to form the set $G_k$. We then use $G_1$ to $G_K$ to fine-tune the text model and generate the fine-tuned question embeddings. Finally, we take the average of the embeddings corresponding to the questions in $G_k$ (Eq. (6)) as the model embedding for LLM$_k$.
>
> For example, assume $n$ (the size of each $Q_m$) is 100. Suppose LLM$_k$ ranks top for 75 questions in $Q_1$, ranks top for 25 questions in $Q_2$, and performs poorly in all other $Q_m$. Then, in Eq. (6), $G_k$ is the union of those 75 + 25 question embeddings, and LLM$_k$ is represented by the averaged embedding of $G_k$.
>
> (lines 270–283) Proposition 1 shows that Eq. (6) performs an implicit categorical reweighting (lines 277 to 279). Continuing the previous example, Proposition 1 states that, in expectation (that is, assuming infinitely many samples for each $Q_m$), the embedding computed by Eq. (6) equals the weighted combination of the expected embeddings ($\mathbb{E}[Q_m]$ in main text (line 274)) of $Q_1$ and $Q_2$, with coefficients $f_{k1}/(f_{k1}+f_{k2})$ and $f_{k2}/(f_{k1}+f_{k2})$. Informally, you can imagine $f_{k1}/(f_{k1}+f_{k2})$  as 0.75/(0.75+0.25) and $f_{k2}/(f_{k1}+f_{k2})$ as 0.25/(0.75+0.25).
>
> We will revise the main text and place this example in the Appendix to improve readability.

---

### Author Response · Authors · 2025-12-03
**Summary of Reviewer-Author Discussion**

Dear AC, SAC, and PC,

We appreciate your efforts and time for reviewing. We prepare a summary of how we have addressed the reviewers’ concerns.

-----

In addition to the detailed responses to each reviewer’s comments, we provide a general response that clarifies the motivation behind our approach and explains key methodological choices. These clarifications address the main concerns raised across the reviews and highlight the contributions and strengths of our work.

Please refer to **General Response and Clarifications: (A) and (B)** and **General Response and Clarifications: (C)**

-----

We have revised the manuscript to address the reviewers’ comments. The modified parts are shown in blue.

- We integrate the **General Response and Clarifications** into the paper.
- We add an example to explain Proposition 1 (lines 882–892).
- In Appendix A.3, we use 2D t-SNE to illustrate the effectiveness of fine-tuning the text embedding model (line 893).
- In Appendix B.5, we report additional experiments on fine-tuning without categorical weighting (line 959 and Figure 8, page 22). These experiments are included to enable a direct comparison with our proposed method, which integrates fine-tuning with categorical weighting, as presented in Section 5.1 (line 336)

-----

We also actively interacted with the reviewers during the discussion period.

For reviewer **hEWF**, we addressed the concerns raised and modified the manuscript to improve readability. We succeeded in obtaining a two-point increase in the rating.

For reviewer **RdwD**, we reported the performance of our two variants (e5b_E4 and mpnet_E4), the optimal online strategy, and a random baseline. Our variants perform close to the optimal strategy, supporting the superiority already shown in the regret curves. We also clarified that Eagle ranks models using empirical scores without exploration, making it unsuitable for handling exploration–exploitation trade-offs or distribution shift, whereas our approach is designed to address these online-adaptivity challenges. Finally, we explained why other baselines are not directly comparable due to differences in supervision type (point-wise vs. pairwise) and routing scope (two models vs. multiple models). Please refer to our detailed reply to reviewer RdwD.

-----

We respectfully invite the AC to consult our updated manuscript, in which all revisions are highlighted in blue. We believe the added discussion, improved illustrations, and new experiments substantially address the reviewers’ major concerns, and we hope these updates will be taken into account in the final evaluation.

Sincerely,

Authors of Submission 16481

---

### Meta-Review · Area_Chair_YQBV · 2026-01-12

**Summary:**

The reviewers' concerns are summarized below:

- The online learning phase introduces significant cost and latency overhead, which seems to contradict the primary motivation of LLM routing, namely cost reduction.

- The assumption that pairwise preference feedback can be synthetically generated via a BTL model is overly idealistic. Consequently, the smooth convergence of the regret curves may be an artifact of this idealized simulation.

- The CCFT method primarily relies on a predefined set of categories. When dealing with cross-domain or novel query types not seen during the offline phase, this rigid structure could lead to suboptimal routing decisions.

- Since prior works have proposed techniques to strengthen users' and models' semantic information, it would be better to discuss why these previous approaches cannot be directly applied or what advantages the proposed CCFT offers.

- The experiments can be further strengthened by including more intuitive evaluation metrics and baselines for an ablation study. The standard deviation is also needed for understanding the performance stability.

- The presentation can still be improved.

- There is a line of prior methods considering model/LLM embedding along with the query embedding for the LLM routing problem which is similar to this work. It would be better to highlight these differences.

- It might be better to include some modern baselines (e.g., Universal routing, CARROT) to showcase the superiority of the proposed methods, even though most of them consider an offline training framework.

- Why did the authors choose to use Thompson sampling for the bandit algorithm part?

- A limitation is that the feedback mechanism is only evaluated in simulation or with automated oracles, rather than with human users.

- The method introduces several moving parts, including fine-tuning embeddings, computing category-weighted representations, and then running a specialized bandit algorithm, which could hinder deployment and are not fully ablated.

- There is no direct comparison to a conventional supervised routing approach or to other online bandit algorithms in the literature.

- The solution is tailored to preference feedback, which is a reasonable design given the motivation, but it does not incorporate pointwise feedback or hybrid approaches.

**Reviewer Concerns:**

I appreciate that the authors have provided a comprehensive response to the reviewers’ concerns, adding further analysis and extensive experiments that have been incorporated into the revised submission. However, I find that some concerns remain only partially addressed, so it cannot be recommended for acceptance in its current version:

- For Reviewer v6zs' concern W1, the authors mention that the regret curve becomes flat as rounds increase, which they interpret as the algorithm choosing the same optimal LLM more often. However, this is only a very qualitative and intuitive argument. From the regret curve, we cannot clearly see how often the optimal LLM is actually chosen, and the curves do not appear particularly flat in the figure. Overall, this is a somewhat vague explanation that is unlikely to convince reviewers. The motivation for applying dueling bandits in LLM routing, especially regarding the cost and latency overhead, needs to be well explained in a more detailed and explicit way, so that the value of this work can be highlighted.

- Reviewer RdwD asked the authors, both in the initial review and during the discussion, to provide accuracy or other more intuitive evaluation metrics beyond cumulative regret. However, the authors only indicated that they were "currently deriving" the accuracy metric and did not explicitly define or report the corresponding results.

- For Reviewer RdwD's concern W2, the authors mentioned that the standard deviation is usually not plotted for a multi-armed bandit method. However, I find such regret curves with error bars can be seen very often in MAB literature, for example, "Neural Dueling Bandits: Preference-Based Optimization with Human Feedback." Thus, the authors need to report the error bar/standard deviation for the regret to address the reviewer's concern.

**Reviewer Scores:**

Reviewers v6zs and hEWF initially gave ratings of 4, while Reviewers RdwD and iqVs gave ratings of 6. After the rebuttal, Reviewer hEWF increased the score from 4 to 6. However, as discussed above, the authors did not fully address the concerns raised by Reviewers v6zs and RdwD. Therefore, I expect Reviewer v6zs to keep the rating at 4, and Reviewer RdwD could at most keep the score of 6 unchanged.

---

### Decision · Program_Chairs · 2026-01-26

Reject